# Pareto-Optimal Fronts for Benchmarking Symbolic Regression Algorithms

**Kei Sen Fong** [1]   **Mehul Motani** [1 2]

## Abstract

Symbolic Regression (SR) algorithms select expressions based on prediction performance while also keeping the expression lengths short to produce explainable white box models. In this context, SR algorithms can be evaluated by measuring the extent to which the expressions discovered are Pareto-optimal, in the sense of having the best R-squared score for a given expression length. This evaluation is most commonly done based on *relative* performance, in the sense that an SR algorithm is judged on whether it Pareto-dominates other SR algorithms selected in the analysis, without any indication on efficiency or attainable limits. In this paper, we explore *absolute* Pareto-optimal (APO) solutions instead, which have the optimal tradeoff between the multiple SR objectives, for 34 datasets in the widely-used SR benchmark, SRBench, by performing exhaustive search. Additionally, we include comparisons between eight numerical optimization methods. We extract, for every dataset, an APO front of expressions that can serve as a universal baseline for SR algorithms that informs researchers of the best attainable performance for selected sizes. The APO fronts provided serves as an important benchmark and performance limit for SR algorithms and is made publicly available at: https://github.com/kentridgeai/SRParetoFronts

## 1. Introduction

Symbolic Regression (SR) is the task of finding closed-form analytical expressions of practical interest that describe the relationship between variables in a measurement dataset.

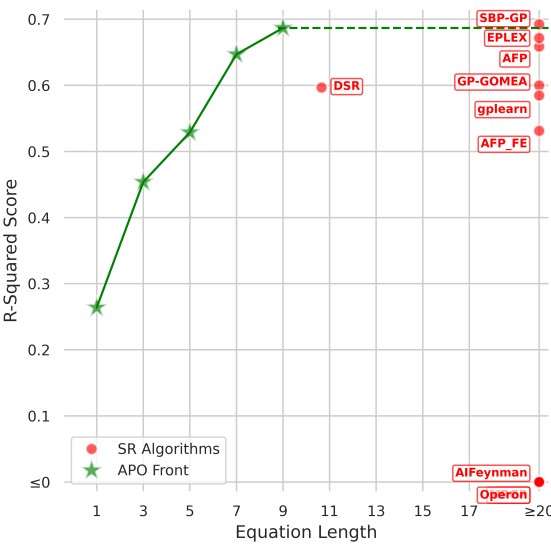

Figure 1: Our APO front on a subset of 34 datasets from SRBench (La Cava et al., 2021). The APO front informs us that the benchmarked SR algorithms may not have sufficiently explored the search space of short expressions in these datasets. Note: Results here are aggregated over the 34 datasets and we perform per-dataset analysis later.

In contrast to black-box machine learning models, SR algorithms produce expressions that are explainable and interpretable. Thus, SR has become a first-class algorithm in various fields including physics (Udrescu & Tegmark, 2020), material sciences (Wang et al., 2019), engineering (Martinez-Gil & Chaves-Gonzalez, 2020) and healthcare (Christensen et al., 2022; Fong & Motani, 2024).

In recent large-scale benchmarking work for SR algorithms, the performance of SR algorithms on black-box regression datasets is measured based on the *relative* performance of the algorithm (La Cava et al., 2021; de Franca et al., 2024). In particular, we refer to a *relative* Pareto-optimal SR algorithm as one that produces expressions with the best R-squared ($R^2$) score for expressions up to a given size, when compared to other SR algorithms selected in the analysis.

In this paper, we propose to move towards *absolute* Pareto-optimality, where we exhaustively search through the en-

[1]Department of Electrical and Computer Engineering, National University of Singapore, Singapore. [2]N.1 Institute for Health, Institute for Digital Medicine (WisDM), Institute of Data Science, National University of Singapore, Singapore. Correspondence to: Kei Sen Fong <fongkeisen@u.nus.edu>, Mehul Motani <motani@nus.edu.sg>.

*Proceedings of the 42nd International Conference on Machine Learning*, Vancouver, Canada. PMLR 267, 2025. Copyright 2025 by the author(s).

tire space of expressions within a maximum expression size to find *absolute* Pareto-optimal (APO) fronts for 34 datasets used in SRBench (La Cava et al., 2021), a widely used SR benchmark. In contrast to synthetic datasets that are generated from a known ground-truth equation, black-box datasets do not possess a baseline equation to compare against. Our APO fronts serve as a useful baseline for benchmarking, and they inform SR researchers about the efficiency and attainable limits of SR algorithms.

To illustrate, Figure 1 shows the performance of SR algorithms from SRBench (in red) and our APO front (in green), averaged across the subset of 34 datasets. The APO front represents a fundamental limit on the performance of any SR algorithm. Figure 1 shows that many algorithms are far from the APO front, performing much worse on both axes and so there is potential room for improvement.

To find these APO fronts, we use an SR algorithm that exhaustively searches expressions of a fixed size. To limit the expression lengths, we adopt gene expression programming (GEP) (Ferreira, 2002) technique of using a genome-phenome system to manipulate and represent expressions. In GEP, the expressions are encoded in their fixed-length *K-expression* form (analogous to genome) and evaluated in their corresponding decoded variable-length mathematical expression form (analogous to phenome). Hence, GEP allows expressions to be easily modified in their fixed-length *K-expression* form, while simultaneously allowing for varying functional complexity (Ferreira, 2002). Other strengths of *K-expressions* include not requiring a validity check since all *K-expressions* are guaranteed to decode into a valid expression. We should, however, note that the entire search space of expressions is large and computationally expensive, in the order of $O(d^l)$, where $d$ is the number of variables in the dataset and $l$ is the expression length. Consequently, our data from exhaustive searches on 34 datasets from SRBench establishes a valuable baseline. Sharing this data will prevent redundant computations and enhance SRBench results with more meaningful insights. These assets are made publicly available at https://github.com/kentridgeai/SRParetoFronts.

The main contributions of this paper are as follows:

1. We extract an APO front of expressions from the exhaustive search for a subset of 34 datasets in the widely-used SR benchmark, SRBench. The APO fronts serve as a useful baseline for benchmarking and inform SR researchers about the efficiency and attainable limits of state-of-the-art SR algorithms.
2. We propose conventions for analyzing SR benchmark results (SRBench) with the APO front. These can help avoid potentially contradictory conclusions in SRBench.
3. We empirically investigate 8 different widely-used numerical optimization methods in obtaining the APO

fronts and compare their performance. This provides SR researchers with large-scale evidence to justify their selection of numerical optimization methods.

## 2. Related Work

### 2.1. SR Benchmarking

SR algorithms have been evaluated for their relative Pareto-dominance in SRBench (La Cava et al., 2021), an important benchmark an SR with extensive hyperparameter search. However, unlike the APO front that we propose to find, the relative Pareto front does not provide any indication of the efficiency and attainable limits of SR. Furthermore, in the relative analysis, the axes do not reflect the true $R^2$ score and true model size. Rather, the axes are the rankings of $R^2$ score and model size, which means that depending on the SR algorithms selected, the conclusion can vary drastically. In this paper, we discover fixed APO fronts that remain as an unchanging universal reference baseline even when new SR algorithms are developed and propose conventions that would enable SR benchmarking insights to be independent of the selection of competing SR algorithms.

### 2.2. Exhaustive Search via *K-expression*

Gene expression programming combines the strengths from genetic algorithms and genetic programming by utilizing a genome-phenome system through the introduction of *K-expressions*. *K-expressions* are strings of fixed-length which are subjected to reproduction (genome). These strings are decoded by forming a variable-length expression from a subset of the string (phenome). The decoding process is done by iterating through the *K-expression* and building an expression tree from top to bottom then from left to right, until no valid symbols can be added. For example, the string '$*+-abcde$' is decoded as $(a+b)*(c-d)$, with the symbol $e$ being in excess and not included in the already full expression. This system enables us to create fixed length strings with expressions of variable size. *K-expressions* are guaranteed to decode to form a valid mathematical expression because they have a tail component, in which only *terminal symbols* are present (Ferreira, 2002). *K-expressions* have been used to develop DistilSR (Fong & Motani, 2023), an SR algorithm. However, in DistilSR, *terminal symbols* are replaced with weighted linear combinations of variables, which consequently exclude certain expression structures. In contrast, our algorithm is better suited to exhaustively cover a larger class of expression structures.

### 2.3. Other Exhaustive Search SR

There also exists other SR algorithms which perform exhaustive search (Kammerer et al., 2020; Bartlett et al., 2023). However, these algorithms tend to make simplifications or

assumptions that have a different search space. In our work, our primary goal is not to develop a state-of-the-art SR algorithm, but rather to extract an APO front of expressions for black-box datasets and an exhaustive search is simply a means to this end. Thus, our algorithm design choice is targeted at using fewer assumptions to provide more robust assets that encompass a larger class of functions.

### 2.4. Numerical Optimization in SR

Numerical optimization has been used in SR to obtain numerical constants in expression (Kommenda et al., 2020; Chen et al., 2015). Instead of using fixed predefined constants in expression trees, the symbol, $\mathbb{R}$, denoting Ephemeral Random Constant (ERC), is introduced. To optimize the values of the ERCs, the Broyden-Fletcher-Goldfarb-Shanno (BFGS) algorithm (Fletcher, 2000) is frequently used for SR algorithms (Biggio et al., 2021; Petersen et al., 2019). In this paper, we empirically investigate various numerical optimization methods in obtaining the APO front for SR, beyond the commonly used ones, and compare their performance. Specifically, we consider the following additional 7 methods: (i) L-BFGS-B (Liu & Nocedal, 1989; Zhu et al., 1997), (ii) conjugate gradient (CG) (Hestenes et al., 1952), (iii) Nelder-Mead (Nelder & Mead, 1965), (iv) Powell (Powell, 1977), (v) sequential least squares programming (SLSQP) (Lawson & Hanson, 1995), (vi) truncated Newton constrained (TNC) (Nash, 2000), (vii) trust-region constrained (trust-constr) (Conn et al., 2000).

## 3. Data Collection and Analysis Conventions

In this section, we describe the procedure used to obtain 34 APO front[1] for 34 datasets in SRBench. This procedure is supplemented by code implementation, that produces the exact same data if the same random seeds are used. In our data collection, randomness is involved because the numerical optimization methods require an initial guess, which is randomly generated. Since this work uses datasets from in SRBench (La Cava et al., 2021), the value of the random seeds used are selected to match that of SRBench. In the final part of the section, we propose conventions for SR analysis that address some issues with current trends in SR benchmarking analysis.

### 3.1. Exhaustive Search Algorithm

In our experiments, we use a primitive function set of {Add,Sub,Mul,Div,Pow}, representing addition, subtraction, multiplication, division and power (the absolute value of the base is taken) respectively, all of which have arity two. The primitive operands comprise of the $d$ features in the dataset, $X$, and the ERC, $\mathbb{R}$. The primitive function set and

the primitive operands set form the *primitive_symbols*. We select two values of *head_length*. Ideally, this value should be higher to evaluate more expressions. However, the number of expressions in the exhaustive search grows exponentially with the *head_length*, which sets practical constraints. Within our high budget of 1,480,000 core-compute-hours, we repeated the search for 10 *random_seed* (11284, 11964, 15795, 21575, 22118, 23654, 29802, 5390, 6265, 860) for *head_length* = 3 and did the search for one *random_seed* (11284) for *head_length* = 4. The random seeds are the same values as the 10 used in SRBench and were used to generate initial guesses for numerical optimization for the range (-1,1). Eight numerical optimization methods (to optimize the ERCs, $\mathbb{R}$) were selected to be evaluated: (i) L-BFGS-B (Liu & Nocedal, 1989; Zhu et al., 1997), (ii) conjugate gradient (CG) (Hestenes et al., 1952), (iii) Nelder-Mead (Nelder & Mead, 1965), (iv) Powell (Powell, 1977), (v) sequential least squares programming (SLSQP) (Lawson & Hanson, 1995), (vi) truncated Newton constrained (TNC) (Nash, 2000), (vii) trust-region constrained (trust-constr) (Conn et al., 2000), (viii) BFGS (Fletcher, 2000). These methods formed the *numerical_optimization_list*. Under these settings, we ensured that the exhaustive search produced exact replicable and reproducible results.

At this point, with sufficient details already presented to the reader, we would like to make a short interjection to add an important caveat that the APO fronts are obtained with respect to a fixed primitive function set and a specific local numerical optimizer. Ideally, the fixed primitive function should include all possible function symbols, and the numerical optimizer should be a true global optimizer that is able to find the best set of numerical constants. In practice, this is not yet possible due to the exponential relationship between the number of primitive functions selected and the lack of a true global optimization algorithm. Starting from this point in the text, it would be more accurate to add the subscript indicating the primitive set used and/or the numerical optimization method used (e.g., $\text{APO}_{(\text{Add, Sub, Mul, Div, Pow}), \text{BFGS}}$). For readability, we add the subscript sparingly, when we feel it is important to remind the reader of this caveat. Finally, it should be noted that despite this caveat, we are already able to find fronts that have a large performance gap with the equations found via existing SR algorithms.

Algorithm 1 outlines the steps for our exhaustive search SR algorithm. At the start of the algorithm, all possible *K-expressions* are constructed. The head component of *K-expressions* is built in Step 3, taking all permutation of primitive symbols (i.e., primitive functions and operands) and each permutation is appended by a tail component (i.e., a chain of terminal symbols) in Step 4 that guarantees that all *K-expressions* generated produce valid expressions (Ferreira, 2002). Here, the length of the tail is determined by $h \times (n_{max} - 1) + 1$, where $h$ is the head length and $n_{max}$ is the

---

[1]Available at https://github.com/kentridgeai/SRParetoFronts

---

**Algorithm 1:** Exhaustive Search Pseudo Code

---

**Input:** *primitive_symbols*, *head_length*, *random_seed*, *numerical_optimization_list*, $X$, where $X$ is an $n \times d$ matrix representing $n$ datapoints with $d$ normalized features (mean 0, standard deviation 1)

**Output:** *search_data*

1   *max_arity* ← MaxOperationArity(*primitive_symbols*)

2   *terminal_symbol_count* ← (*head_length* × (*max_arity* − 1) + 1)

3   *population* ← Permutation(*primitive_symbols*, *head_length*)

     /* Tail component of terminal symbols ensures decodablility           */

4   *population* ← AppendTerminalSymbols(*population*, *terminal_symbol_count*)

5   *search_data* ← *null*

6   **for** Method ∈ *numerical_optimization_list* **do**

7      **for** *K-expression* ∈ *population* **do**

8          *expression* ← Decode(*K-expression*)

           /* *random_seed* is required to ensure reproducibility of initial guess

              of numerical parameters used in numerical optimization        */

9          *expression* ← Method(*expression*, $X$, *random_seed*)

10        *expression_score* ← RSquaredScore(*expression*, $X$)

11        *search_data* ← *search_data* ∪ {(*expression*, *expression_score*)}

12     **end**

13   **end**

14   **return** *search_data*

---

maximum operand arity of the primitive function set. Then, we iterate through all eight numerical optimization methods (see Step 6) and *K-expressions* (see Step 7) and perform numerical optimization. The optimized expression and its $R^2$ score on the dataset $X$ are then stored. This forms the raw data which we make available. For *head_length* = 3, we ran Algorithm 1 on 34 datasets from SRBench (see Appendix A for dataset details), which had the condition that there were less than 1000 datapoints and less than 10 features, and for *head_length* = 4, we reduced this to 30 datasets by excluding datasets with more than 6 features.

### 3.2. Extracting APO Front

For each dataset, we extract the APO front for the dataset by taking the expressions with the highest $R^2$ score among all the random seeds and all the numerical optimization methods used. One expression for each unique expression length is selected. We set a condition that expressions of larger length must perform better than the expressions of smaller length to qualify for inclusion into the APO front, otherwise no expressions are selected for that length. In Table 1, we show the exact expressions on the APO front for 9 datasets. The table for all 34 datasets is included in Appendix B. The expressions in these Tables are the most important main asset of this paper.

### 3.3. Proposed Conventions for SR Benchmarking

Current practices in SR benchmarking have been useful in summarizing the relative performances of various SR algorithms. Common practices used, which we discuss below, can help to create visually informative plots. In an effort to further improve these practices, particularly in Pareto analysis, we propose two conventions as potential areas of improvement.

#### 3.3.1. CONVENTION #1: FOR PARETO ANALYSIS IN SR BENCHMARKING, CONSIDER USING THE ACTUAL QUANTITY AS AXES.

In current and recent SR Pareto analysis, the axes do not reflect the true $R^2$ score and true model size but instead reflect the rankings of $R^2$ score and model size. Depending on the set of SR algorithms selected, the conclusion can vary drastically, such as gplearn's performance shown in Figure 2a & 2b. Additionally, ranking removes substantial important information. For example, based on Figure 2a, gplearn seem to be very close to DSR in terms of expression length, but in terms of the actual values in Figure 2c, this is not the case, with gplearn having more than three times the length of DSR and is in fact very close in length to BSR, which could not have been inferred from Figure 2a. Furthermore, Pareto analysis in other fields is more commonly in terms of the actual quantity and not rankings of the quantity.

A phenomenon, which we term as Rank Inversion Paradox (see Appendix M for worked examples and elaboration), can also occur, causing contradictory conclusions when rankings are involved in Pareto analysis.

It is understood that using ranked axes can provide better spaced datapoints which can convey messages more

Table 1: APO Front Expressions for 9 Datasets (Full table for all 34 datasets in Appendix B).

| Dataset | APO Front Expressions (in increasing length, separated by '→') |
|---|---|
| 1027_ESL | $x_2$→Mul($x_2$,0.823)→Mul(0.541,Add($x_3$,$x_1$))→Mul(Add(Add($x_3$,$x_2$),$x_1$),0.366) →Sub(Mul(Add($x_2$,$x_1$),0.35),Mul($x_3$,-0.399)) |
| 1096_FacultySalaries | $x_3$→Mul($x_3$,0.926)→Sub(Mul($x_1$,1.89),$x_2$)→Add(Mul($x_2$,-0.966),Mul($x_1$,1.85)) →Div(Sub(Mul($x_1$,1.84),$x_2$),Pow($x_2$,-0.189) |
| 192_vineyard | $x_0$→Mul($x_0$,0.747)→Sub(1.29,Pow(-0.475,$x_0$))→Div($x_0$,Add(Pow($x_0$,$x_1$),0.313)) →Mul(Add($x_1$,-2.24),Sub(Pow(-0.712,$x_0$),1.16) |
| 210_cloud | $x_3$→Mul($x_3$,0.858)→Mul(0.516,Add($x_3$,$x_2$))→Sub(Mul($x_3$,0.55),Mul($x_2$,-0.483)) →Sub(Mul(Add($x_3$,$x_2$),0.617),Mul($x_4$,0.214)) |
| 228_elusage | -1.66e-09→Mul($x_0$,-0.883)→Add(Pow(-0.484,$x_0$),-1.29) →Add(-1.49,Div(Pow(-0.532,$x_0$),0.817))→Sub(Add(Pow(0.48,$x_0$),-1.32),Div(-0.00454,$x_0$) |
| 230_machine_cpu | $x_2$→Mul($x_2$,0.863)→Div($x_2$,Pow($x_3$,-0.325))→Sub(Pow(1.48,$x_2$),Pow(0.656,$x_3$)) →Sub(Sub(Pow(x[0],$x_2$),x[1]),Mul($x_3$,x[2]) |
| 485_analcatdata_vehicle | 1.55e-15→Mul($x_0$,-0.746)→Sub(Mul($x_2$,-0.287),$x_0$) →Add(Mul($x_2$,-0.287),Mul($x_0$,-0.746))→Add(Div(-0.746,$x_0$),Mul(-0.245,Sub($x_2$,$x_1$)) |
| 519_vinnie | $x_1$→Mul($x_1$,0.867)→Mul(-1.84,Mul($x_1$,-0.472)) →Div(-25.5,Sub(Div(-26.8,$x_1$),$x_1$))→Div(-25.5,Sub(Div(-26.8,$x_1$),$x_1$) |
| 522_pm10 | 8.23e-10→Mul($x_0$,0.356)→Mul(0.308,Sub($x_0$,$x_2$)) →Sub(Pow(-1.59,$x_0$),Pow($x_3$,-0.169)) |

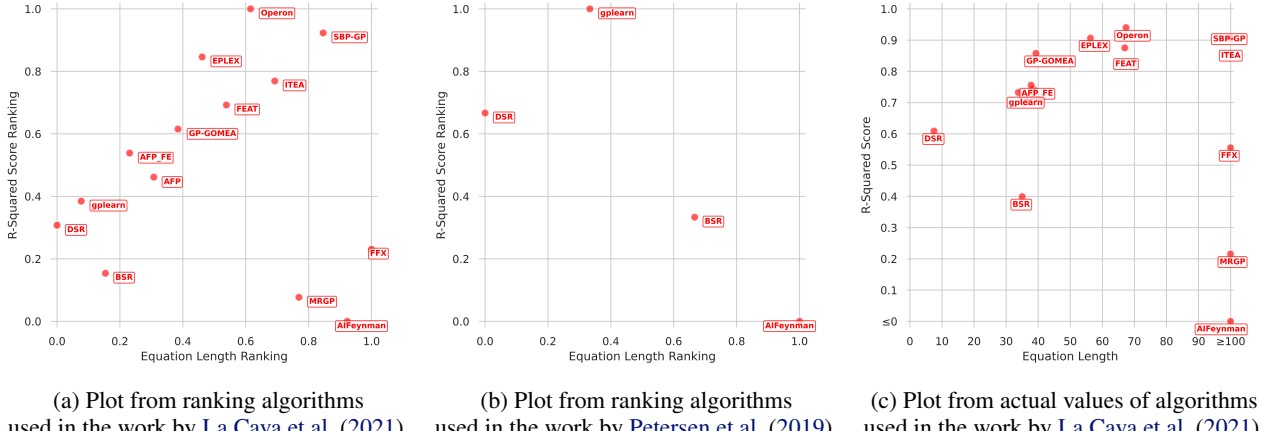

(a) Plot from ranking algorithms used in the work by La Cava et al. (2021).

(b) Plot from ranking algorithms used in the work by Petersen et al. (2019).

(c) Plot from actual values of algorithms used in the work by La Cava et al. (2021).

Figure 2: Three graphs based on the same data (dataset 579_fri_c0_250_5), but with very different visual conclusions. For transferable results across papers, actual values are preferred over ranks. See Appendix M for inversion paradox with ranks.

efficiently, but the cost of potentially conveying misleading conclusions is high. We hope the trend of using only rankings in SR Pareto analysis can be reconsidered and supplemented with analysis using actual quantities.

Finally, presenting results in actual values also has the additional benefit of making the plots and conclusions transferable to other SR works.

3.3.2. CONVENTION #2: FOR PARETO ANALYSIS IN SR BENCHMARKING, CONSIDER SUPPLEMENTARY ANALYSIS WITH PER-DATASET RESULTS TO CONFIRM THE TREND.

In typical SR analysis, results are often aggregated across datasets, while trends on individual datasets are omitted in the work. By aggregating results, too much simplifica-

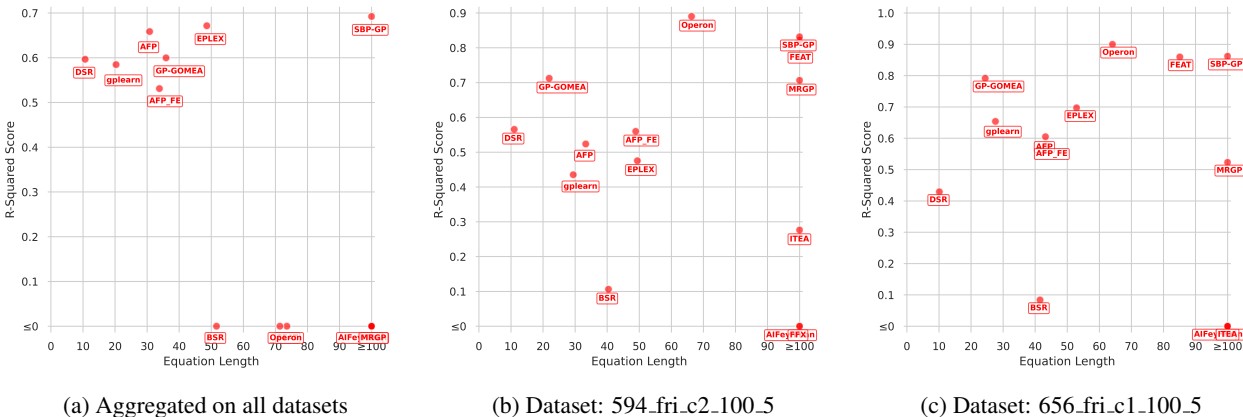

(a) Aggregated on all datasets      (b) Dataset: 594_fri_c2_100_5      (c) Dataset: 656_fri_c1_100_5

Figure 3: By aggregating results, information is lost. For example, Operon's good performance on individual datasets is not communicated to researchers via the aggregated plot.

tion may have occurred, and information is lost. In Figure 3, it appears that DSR, AFP, EPLEX and SBP-GP are on the *relative* Pareto-optimal front from the aggregated results. However, it turns out that for the methods DSR, gplearn, GP-GOMEA, Operon, AFP_FE, AFP, FEAT, EPLEX, SBP-GP, FFX, AIFeynman, their respectively frequency of being on the *relative* Pareto-optimal front across individual datasets are 100%, 67%, 63%, 50%, 37%, 33%, 30%, 27%, 17%, 10%, 3%, respectively. Note that contrary to aggregate results, AFP, EPLEX and SBP-GP are not on the APO front for most datasets. To better represent results, we recommend providing supplementary Pareto analysis conducted on individual datasets.

## 4. Results and Discussion

In this section, we analyze the results obtained from our collected data and extracted APO fronts. We compare against the 14 SR methods benchmarked in SRBench (La Cava et al., 2021), under GPL-3.0 license: Age-Fitness Pareto Optimization (AFP) (Schmidt & Lipson, 2010), Age-Fitness Pareto Optimization with Co-evolved Fitness Predictors (AFP_FE) (Schmidt & Lipson, 2010), AIFeynman (Udrescu et al., 2020), Bayesian Symbolic Regression (BSR) (Jin et al., 2019), Deep Symbolic Regression (DSR) (Petersen et al., 2019), Epsilon-Lexicase Selection (EPLEX) (La Cava et al., 2016), Feature Engineering Automation Tool (FEAT) (La Cava et al., 2018), Fast Function Extraction (FFX) (Mc-Conaghy, 2011), Genetic Programming-based Gene-pool Optimal Mixing Evolutionary Algorithm (GP-GOMEA) (Virgolin et al., 2017), Interaction-Transformation Evolutionary Algorithm (ITEA) (de Franca & Aldeia, 2021), Multiple Regression Genetic Programming (MRGP) (Arnaldo et al., 2014), Operon (Burlacu et al., 2020), Semantic

Backpropagation Genetic Programming (SBP-GP) (Virgolin et al., 2019), gplearn (Stephens, 2016).

### 4.1. Finding #1: Are current methods benchmarked in SRBench close to the APO front?

Based on Figure 1, it seems as though none of the methods are close to the $\text{APO}_{(\text{Add, Sub, Mul, Div, Pow})}$ front, and only SBP-GP has an $R^2$ score that exceeds that of the APO front, but at the cost of a much higher expression length. Looking at individual datasets, in Figure 4, we show examples of individual APO fronts and classify each into one of 4 types (the characteristic of each type is described in the captions). Among the 34 APO fronts, 15 fall under Type I, 5 fall under Type II, 5 fall under Type III and 9 fall under Type IV (see Appendix C for exact details). The performance varies largely across different datasets. Notably, for datasets that fall under Type IV, all SR algorithms fail to sufficiently search the space of small expressions, otherwise, they would have performance close to the APO front. It is, however, promising to see DSR consistently perform relatively close to the APO front consistently, as shown in the examples in Figures 4a, 4b, 4c.

### 4.2. Finding #2: How rare is it to obtain an expression with performance close to the APO front?

In most datasets, it is rare to obtain an expression with performance close to the $\text{APO}_{(\text{Add, Sub, Mul, Div, Pow})}$ front. In Figure 5, we show selected examples of the distribution of $R^2$ score in the search space. These three examples (Figures 5a, 5b, 5c) were selected as their top-bin in the histogram had the minimum, median and maximum value among all other histograms. Additionally, among all the histograms, 85.3% have less than 1% within the top-bin in

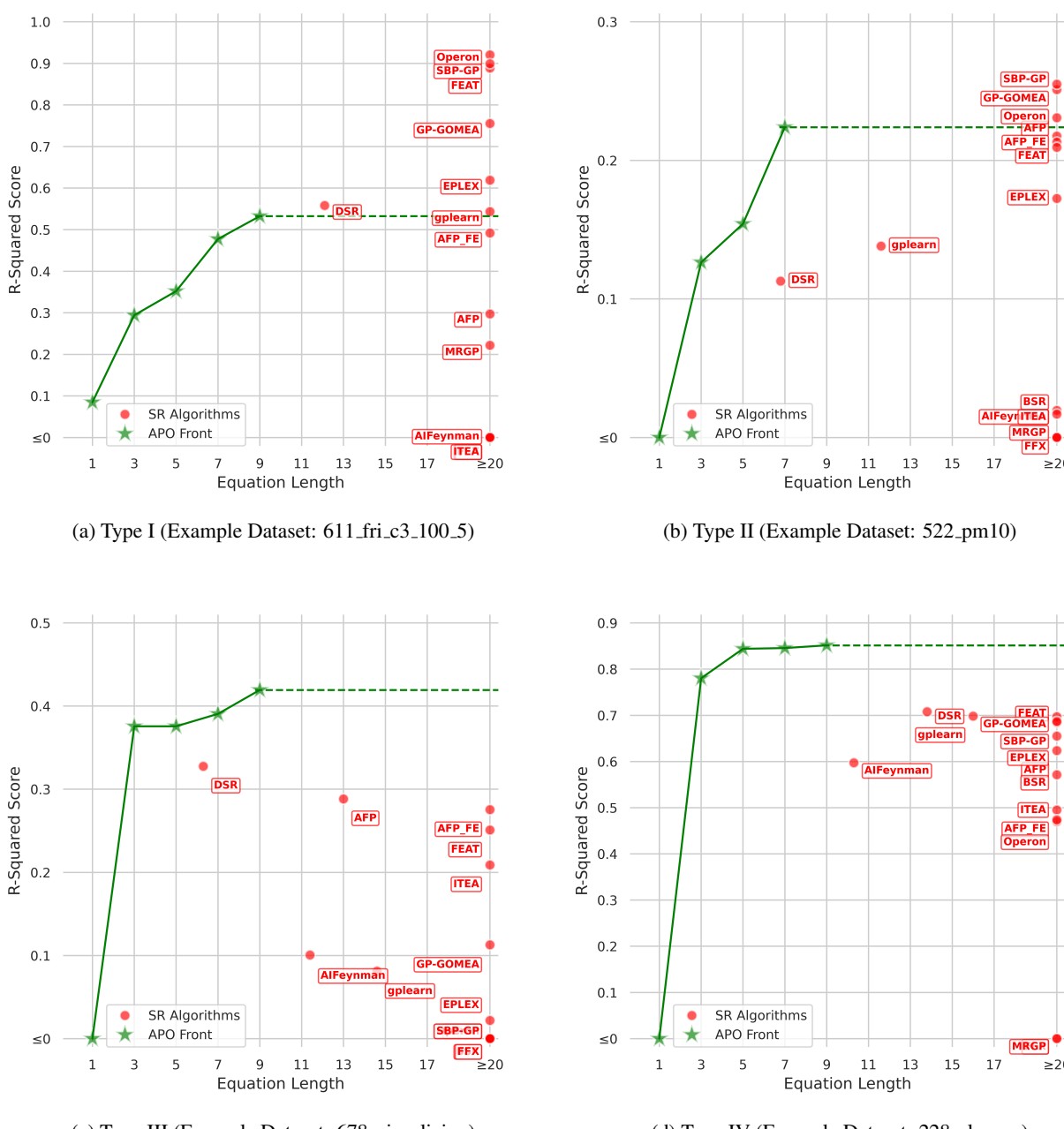

Figure 4: Examples of APO fronts for the 34 datasets. Type I refers to cases where there are SR algorithms of larger length that exceed the $R^2$ score of the APO front. Type II is the same as Type I, but the SR algorithms do not exceed by more than 0.1 $R^2$ score difference. Type III refers to cases where there are no SR algorithms of larger length that exceed the $R^2$ score of the APO front, but there is at least one SR algorithm that is less than 0.1 $R^2$ score away. Type IV is the same as Type III, but with all SR algorithms more than 0.1 $R^2$ score away. The dotted lines are extensions of the longest APO front expression we could attain due to realistic computation constraints in exhaustively searching larger expressions.

the histogram, 66.9% have less than 0.1% within the top-bin in the histogram, 35.2% have less than 0.01% within the top-bin in the histogram. This tells us that a simple random sampling of expressions from the exhaustive search space of short expressions is unlikely to provide performance close to the APO front.

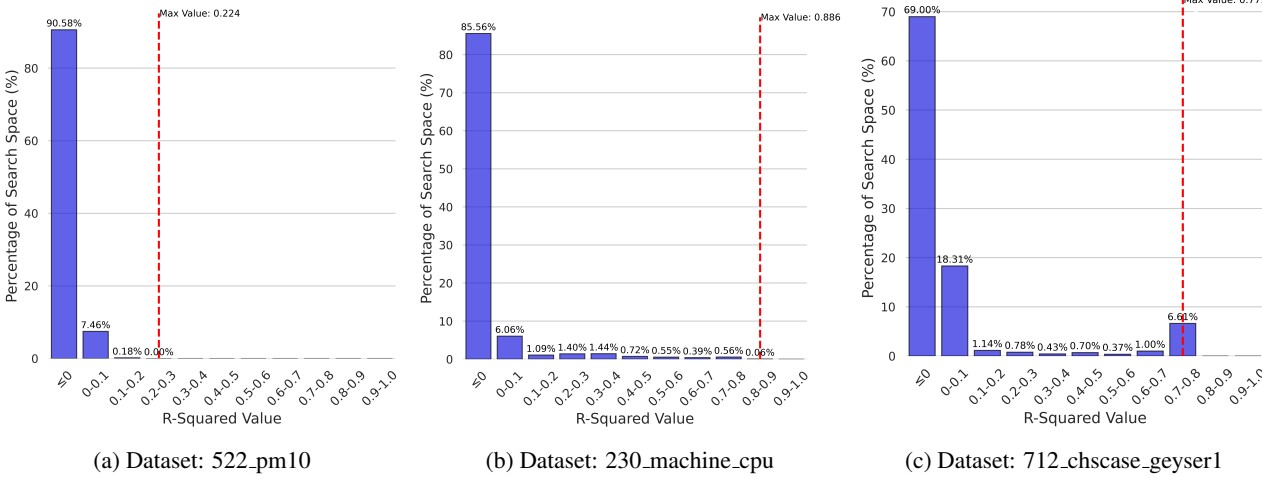

(a) Dataset: 522_pm10       (b) Dataset: 230_machine_cpu      (c) Dataset: 712_chscase_geyser1

Figure 5: Histograms of 3 examples of the distribution of $R^2$ score among the exhaustive search space of expressions. The left graph's top-bin contains 0.0002% (the lowest among all experiments), the middle graph's top bin contains 0.06% (the median) and the right graph's top-bin contains 6.61% (the most).

### 4.3. Finding #3: How is the loss landscape, in terms of expression structure around expressions in the APO front, like?

Unlike the loss landscape on numerical parameters, visualizing the loss landscape in terms of expression structure is challenging because there is no 'nice' continuous axis to represent changes in structure. To this end, we describe the loss landscape by considering if expressions that are two-mutations away from the APO front are likely to mutate back into the APO front.

To measure this, we take all expressions that are two-mutations away from the APO front and apply two greedy one-step mutations. To illustrate this, let us consider the expression Sub(Mul($x_3$,0.55),Mul($x_1$,-0.531)) (on the APO front for dataset 1027_ESL). We generate expressions two-mutations away structurally from this expression. One example is Div(Pow(xdata[3],8.74),Mul(xdata[1],63600)), which then considers all one-step mutations and greedily picks the best to mutate into Div(Mul(xdata[3],-0.0326),Mul(xdata[1],1.16)), which in turn considers all one-step mutations and greedily picks the best to mutate into Sub(Mul($x_3$,0.55),Mul($x_1$,-0.531)). In this case, the initial expression that is two-mutations away from the APO front is able to greedily mutate back to the APO front.

Another example is Mul(Div($x_3$,-0.658),Mul($x_1$,0.0416)), which then considers all one-step mutations and greedily picks the best to mutate into Mul(Div($x_3$,-240),Sub($x_1$,180)), which in turn considers all one-step mutations and greedily picks the best to mutate into Mul(Mul($x_3$,-3.37e-3),Sub($x_1$,223)). In this case, the initial expression that is two-mutations away from the APO front is unable to greedily mutate back to the APO front.

On average across the datasets, 23.7% of expressions that are two-mutations away from the APO front are able to greedily mutate back to the APO front. Dataset 659_sleuth_ex1714 had the lowest rate of 0% and dataset 690_visualizing_galaxy had the highest rate of 68.8%. Our collected data provides us with a powerful database that allows us to analyze the loss landscape in terms of expression structure, which has hardly been analyzed in SR. Further metrics can be created and analyzed in depth (such as changing two-mutations to k-mutations), but this requires a thorough separate analysis and far extends beyond the scope of this paper, so we leave extensions of this idea to future works in SR.

### 4.4. Finding #4: How does the choice of numerical optimization affect results?

To answer this, we compare the various APO fronts (i.e., comparing APO$_{\text{(Add, Sub, Mul, Div, Pow), Powell}}$, APO$_{\text{(Add, Sub, Mul, Div, Pow), BFGS}}$, APO$_{\text{(Add, Sub, Mul, Div, Pow), TNC}}$ and 5 other APO fronts from the other numerical optimization methods). Empirically, there is not a significant difference in the results across different numerical optimization methods as evidenced in Figure 6, where we display the histograms with the largest difference. In terms of KL divergence, Figures 6a & 6b showed the largest difference, followed by Figures 6a & 6c. Yet, the distribution of $R^2$ score in the search space appears largely similar despite being the ones with the largest difference. We also consider only the expressions on the APO front, and evaluate the standard deviation of the $R^2$ score of these expressions across varying numerical optimization methods. The mean standard deviation of $R^2$ score ranged from a minimum of 0.0 to a maximum of 0.043, with a mean and median

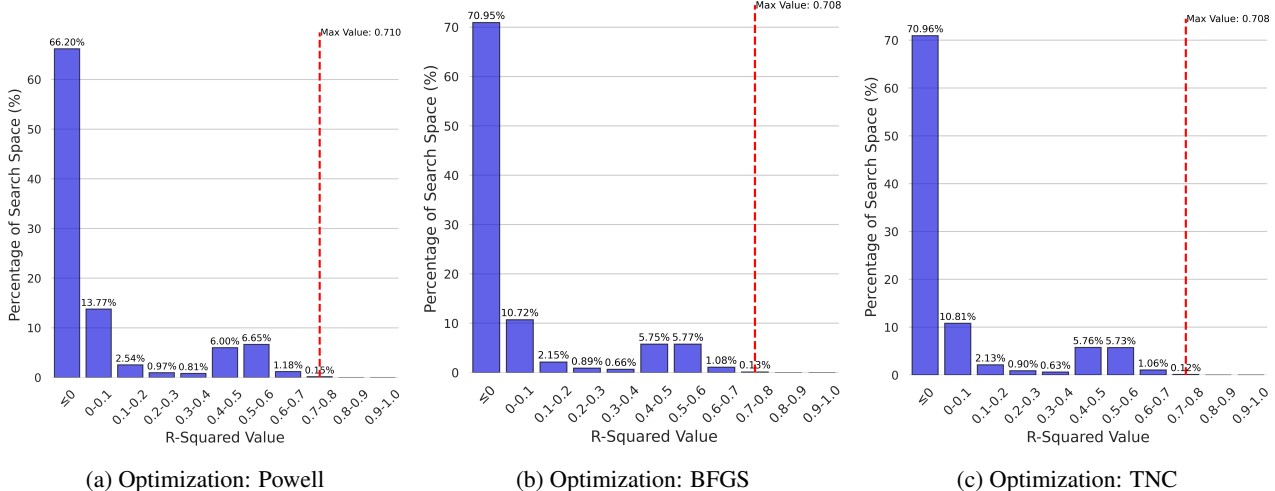

(a) Optimization: Powell      (b) Optimization: BFGS      (c) Optimization: TNC

Figure 6: Histograms of the distribution of $R^2$ score across different numerical optimization methods on the dataset 192_vineyard. We chose to display this dataset and these three optimization methods because the difference in probability distributions (measured via KL divergence) is the greatest here (among all other numerical optimization and datasets).

of 0.00043 and 3.42e-19 respectively. Only 1.47% of the standard deviation measured exceeded 0.01. These observations indicate the expressions are stable with respect to the numerical optimization method selected and in terms of $R^2$ score, there is no strong reason to favor one more than another. Though the differences between the numerical optimization methods are small, for completeness of results, the percentage of times in which each method produces the expression on the APO front are 32.1%, 25.0%, 20.0%, 10.9%, 9.41%, 2.06%, 0.588% and 0.00% for the methods Powell, CG, trust-constr, L-BFGS-B, BFGS, TNC, SLSQP and Nelder Mead respectively. The commonly used BFGS does not have a clear advantage in obtaining expressions on the APO front compared to the other methods.

### 4.5. Finding #5: Which datasets should SR focus on?

Based on the discussion earlier in Finding #1 and Figure 4, for datasets with results under Type III and Type IV, current SR algorithms are not sufficiently searching the space of short expressions, which SR researchers should focus on. Even for datasets with results under Type II, the performance of short expressions on our APO$_{(Add, Sub, Mul, Div, Pow)}$ front are comparable with SR algorithms performance, which demands investigation.

### 4.6. Limitations and Societal Impact

In the most precise and definite sense of word 'absolute', the numerical optimization method needs to provide the global minimum. However, there are no numerical optimization methods that have this property. To help mitigate this, we ran experiments over different initial guesses (set by the random seed) for numerical optimization. We found that

the prediction performance of APO fronts is similar across different random seeds (the standard deviation for expressions across the random seeds is less than 0.1 for 86.2% of the expressions), which gives us some confidence that the parameters are not often stuck at local optima, otherwise the prediction performance would have been different. We also could not include all possible operators. Finally, though the computation resources spent on the project is high, we hope that by making these data publicly available, it reduces the computational burden on other researchers and allows researchers to focus on new discoveries without the need for redundant computational efforts.

### 5. Conclusion

In this work, we move towards absolute Pareto optimal fronts, which enables SR researchers to determine the absolute difference between SR algorithms and the best attainable expressions. This complements current SR benchmarking efforts in which the relative performance with other SR algorithms is made instead. We also propose conventions to improve upon common practices in SR benchmarking efforts to reduce the tendency for misleading conclusions to be drawn. Then, we compare among different numerical optimization methods in obtaining the APO fronts to investigate the choice of numerical optimization method to use in SR. Finally, we report on several findings that can potentially help SR algorithm design. A key finding is that the search space of short, simple equations is sufficiently expressive and should be explored more before expanding the search space to longer equations, a mechanism that is related to increasing explainability, which is a primary reason for practitioners to pick SR over alternative machine learning algorithms in the first place.

## Acknowledgements

This research/project is supported by the National Research Foundation, Singapore under its AI Singapore Programme (AISG Award No: AISG3-PhD-2023-08-052T), and A*STAR, CISCO Systems (USA) Pte. Ltd and National University of Singapore under its Cisco-NUS Accelerated Digital Economy Corporate Laboratory (Award I21001E0002). The computational work for this article was partially performed on resources of the National Supercomputing Centre, Singapore (https://www.nscc.sg).

## Impact Statement

This paper presents work whose goal is to advance the field of Machine Learning. There are many potential societal consequences of our work, none which we feel must be specifically highlighted here.

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

# A. More Details on the 34 Selected Datasets

Table 2: Details of the subset of the 34 selected datasets from SRBench.

| Dataset | Number of Datapoints | Number of Features |
|---|---|---|
| 1027_ESL | 488 | 4 |
| 1096_FacultySalaries | 50 | 4 |
| 192_vineyard | 52 | 2 |
| 210_cloud | 108 | 5 |
| 228_elusage | 55 | 2 |
| 230_machine_cpu | 209 | 6 |
| 485_analcatdata_vehicle | 48 | 4 |
| 519_vinnie | 380 | 2 |
| 522_pm10 | 500 | 7 |
| 523_analcatdata_neavote | 100 | 2 |
| 547_no2 | 500 | 7 |
| 556_analcatdata_apnea2 | 475 | 3 |
| 557_analcatdata_apnea1 | 475 | 3 |
| 561_cpu | 209 | 7 |
| 579_fri_c0_250_5 | 250 | 5 |
| 594_fri_c2_100_5 | 100 | 5 |
| 596_fri_c2_250_5 | 250 | 5 |
| 597_fri_c2_500_5 | 500 | 5 |
| 601_fri_c1_250_5 | 250 | 5 |
| 611_fri_c3_100_5 | 100 | 5 |
| 613_fri_c3_250_5 | 250 | 5 |
| 617_fri_c3_500_5 | 500 | 5 |
| 624_fri_c0_100_5 | 100 | 5 |
| 631_fri_c1_500_5 | 500 | 5 |
| 649_fri_c0_500_5 | 500 | 5 |
| 656_fri_c1_100_5 | 100 | 5 |
| 659_sleuth_ex1714 | 47 | 7 |
| 663_rabe_266 | 120 | 2 |
| 665_sleuth_case2002 | 147 | 6 |
| 678_visualizing_environmental | 111 | 3 |
| 687_sleuth_ex1605 | 62 | 5 |
| 690_visualizing_galaxy | 323 | 4 |
| 706_sleuth_case1202 | 93 | 6 |
| 712_chscase_geyser1 | 222 | 2 |

Note: The Friedman datasets (Friedman, 2001) are synthetically generated by summing a collection of smooth, bell-shaped components, where each component is given a randomly chosen scale and depends only on a small, randomly selected subset of the inputs. Each component is centered at a random point in the input space and its shape is determined by picking a random orientation and applying random stretch factors along each direction. Inputs themselves are drawn from a standard normal distribution. For finer details of the process, please refer to the original work by Friedman (2001).

## B. APO Front Expressions for All Datasets

Table 3: APO Front Expressions for all 34 Datasets.

| Dataset | APO Front Expressions (in increasing length, separated by '→') |
|---|---|
| 1027_ESL | $x_2$→Mul($x_2$,0.823)→Mul(0.541,Add($x_3$,$x_1$))→Mul(Add(Add($x_3$,$x_2$),$x_1$),0.366) →Sub(Mul(Add($x_2$,$x_1$),0.35),Mul($x_3$,-0.399)) |
| 1096_FacultySalaries | $x_3$→Mul($x_3$,0.926)→Sub(Mul($x_1$,1.89),$x_2$)→Add(Mul($x_2$,-0.966),Mul($x_1$,1.85)) →Div(Sub(Mul($x_1$,1.84),$x_2$),Pow($x_2$,-0.189)) |
| 192_vineyard | $x_0$→Mul($x_0$,0.747)→Sub(1.29,Pow(-0.475,$x_0$))→Div($x_0$,Add(Pow($x_0$,$x_1$),0.313)) →Mul(Add($x_1$,-2.24),Sub(Pow(-0.712,$x_0$),1.16) |
| 210_cloud | $x_3$→Mul($x_3$,0.858)→Mul(0.516,Add($x_3$,$x_2$))→Sub(Mul($x_3$,0.55),Mul($x_2$,-0.483)) →Sub(Mul(Add($x_3$,$x_2$),0.617),Mul($x_4$,0.214)) |
| 228_elusage | -1.66e-09→Mul($x_0$,-0.883)→Add(Pow(-0.484,$x_0$),-1.29) →Add(-1.49,Div(Pow(-0.532,$x_0$),0.817))→Sub(Add(Pow(0.48,$x_0$),-1.32),Div(-0.00454,$x_0$)) |
| 230_machine_cpu | $x_2$→Mul($x_2$,0.863)→Div($x_2$,Pow($x_3$,-0.325))→Sub(Pow(1.48,$x_2$),Pow(0.656,$x_3$)) →Sub(Sub(Pow(x[0],$x_2$),x[1]),Mul($x_3$,x[2]) |
| 485_analcatdata_vehicle | 1.55e-15→Mul($x_0$,-0.746)→Sub(Mul($x_2$,-0.287),$x_0$) →Add(Mul($x_2$,-0.287),Mul($x_0$,-0.746))→Add(Div(-0.746,$x_0$),Mul(-0.245,Sub($x_2$,$x_1$)) |
| 519_vinnie | $x_1$→Mul($x_1$,0.867)→Mul(-1.84,Mul($x_1$,-0.472)) →Div(-25.5,Sub(Div(-26.8,$x_1$),$x_1$))→Div(-25.5,Sub(Div(-26.8,$x_1$),$x_1$) |
| 522_pm10 | 8.23e-10→Mul($x_0$,0.356)→Mul(0.308,Sub($x_0$,$x_2$)) →Sub(Pow(-1.59,$x_0$),Pow($x_3$,-0.169)) |
| 523_analcatdata_neavote | -2.13e-09→Mul($x_0$,-0.969)→Add(-1.33,Pow($x_0$,9.23)) →Add(Pow(Sub($x_0$,0.251),2.75),-1.21)→Sub(Sub(Mul($x_0$,-1.08),0.00779),Div(-0.113,$x_0$)) |
| 547_no2 | $x_0$→Mul($x_0$,0.512)→Mul(0.466,Sub($x_0$,$x_2$)) →Sub(Mul($x_2$,-0.382),Mul($x_0$,-0.549)) |
| 556_analcatdata_apnea2 | -3.71e-09→Mul($x_0$,0.0917)→Div(0.0864,Sub($x_0$,0.599)) →Div(-0.0492,Sub(Pow(-0.597,$x_0$),$x_1$))→Div(-0.0492,Sub(Pow(-0.597,$x_0$),$x_1$) |
| 557_analcatdata_apnea1 | 1.17e-09→Mul($x_1$,0.106)→Div(0.0889,Sub($x_1$,0.596)) →Div(-0.0532,Sub(Pow(-0.596,$x_1$),$x_0$))→Div(-0.0532,Sub(Pow(-0.596,$x_1$),$x_0$) |
| 561_cpu | $x_3$→Mul($x_3$,0.901)→Div($x_3$,Pow($x_4$,-0.355)) →Sub(Pow(-1.5,$x_3$),Pow(-0.703,$x_4$) |
| 579_fri_c0_250_5 | $x_3$→Mul($x_3$,0.546)→Mul(0.5,Add($x_3$,$x_0$))→Mul(Add(Add($x_1$,$x_0$),$x_3$),0.466) →Add(Mul(Add($x_1$,$x_0$),0.397),Mul($x_3$,0.615)) |
| 594_fri_c2_100_5 | 1.8e-16→Mul($x_1$,-0.403)→Div($x_0$,Sub($x_4$,2.79)) →Sub(1.14,Pow(Div(-2.03,$x_0$),$x_0$))→Mul(Div($x_0$,-0.704),Pow(Add($x_0$,-1.9),$x_1$) |
| 596_fri_c2_250_5 | -2.24e-09→Mul($x_1$,-0.404)→Mul(0.355,Sub($x_3$,$x_1$))→Sub(1.17,Pow(Div(-1.78,$x_1$),$x_1$)) →Mul(Div($x_1$,x[0]),Pow(Sub($x_1$,x[1]),$x_0$) |
| 597_fri_c2_500_5 | -1.05e-09→Mul($x_1$,-0.387)→Mul(0.347,Sub($x_3$,$x_1$))→Sub(1.13,Pow(Div(-1.84,$x_0$),$x_0$)) →Sub(Mul(Pow($x_1$,1.77),$x_0$),Add($x_1$,$x_0$) |
| 601_fri_c1_250_5 | -2.99e-09→Mul($x_3$,0.383)→Mul(-0.359,Sub($x_1$,$x_3$))→Mul(Add(Pow($x_1$,1.6),-1.93),$x_1$) →Sub(Pow(Div($x_1$,-1.64),$x_1$),Pow(-0.719,$x_3$) |
| 611_fri_c3_100_5 | $x_3$→Mul($x_3$,0.542)→Div($x_3$,Sub(2.41,$x_1$))→Sub(Pow($x_1$,$x_1$),Pow(-0.632,$x_3$)) →Sub(Pow(Div($x_1$,-2.62),$x_1$),Sub(1.33,$x_4$)) |
| 613_fri_c3_250_5 | -2.62e-09→Div($x_3$,2)→Div($x_3$,Sub(2.59,$x_1$))→Mul(Add(Mul($x_1$,$x_0$),-1.82),$x_1$) →Sub(Pow(Mul($x_0$,-0.696),$x_0$),Pow(-0.652,$x_3$)) |
| 617_fri_c3_500_5 | 1.55e-15→Mul($x_3$,0.406)→Div($x_3$,Sub(2.88,$x_1$))→Add(-0.755,Pow(Sub($x_0$,0.554),$x_1$)) →Sub(Pow(Mul($x_1$,0.69),$x_1$),Pow(0.692,$x_3$)) |
| 624_fri_c0_100_5 | $x_1$→Mul($x_1$,0.53)→Mul(0.497,Add($x_3$,$x_1$))→Mul(Add(Add($x_3$,$x_1$),$x_0$),0.467) →Add(Mul(Add($x_3$,$x_1$),0.497),Mul($x_0$,0.408)) |
| 631_fri_c1_500_5 | 1.11e-16→Mul($x_3$,0.395)→Mul(-0.355,Sub($x_1$,$x_3$))→Mul(Add(Pow($x_1$,1.41),-1.83),$x_1$) →Sub(Pow(Div($x_1$,-1.64),$x_1$),Pow(-0.679,$x_3$)) |
| 649_fri_c0_500_5 | $x_3$→Mul($x_3$,0.61)→Mul(0.494,Add($x_3$,$x_0$))→Mul(Add(Add($x_3$,$x_1$),$x_0$),0.464) →Sub(Mul(Add($x_1$,$x_0$),0.392),Mul($x_3$,-0.594)) |
| 656_fri_c1_100_5 | -4.72e-09→Mul($x_3$,0.387)→Div(x[0],Sub($x_0$,x[1]))→Add(-0.46,Pow(Pow(-0.0895,$x_0$),$x_1$)) →Sub(Pow(Add($x_1$,-0.755),$x_0$),Pow($x_1$,0.737)) |
| 659_sleuth_ex1714 | $x_5$→Mul($x_5$,0.891)→Mul(Pow($x_0$,0.293),$x_5$) →Div(Mul($x_5$,5.54),Sub(6.84,$x_2$)) |
| 663_rabe_266 | $x_0$→Mul($x_0$,0.938)→Sub($x_0$,Mul($x_1$,0.304))→Add(Add($x_0$,-1.05),Pow(-0.731,$x_1$)) →Add(Add($x_0$,-0.395),Div(Pow(-0.414,$x_1$),3.64)) |
| 665_sleuth_case2002 | $x_5$→Mul($x_5$,0.585)→Sub(1.14,Pow(0.613,$x_5$))→Div(Add(Pow(-0.362,$x_5$),-1.8),-3.56) →Sub(Pow(Sub($x_5$,-0.998),-0.197),Pow(0.607,$x_5$)) |
| 678_visualizing_environmental | 6.18e-09→Mul($x_0$,-0.613)→Mul(Div($x_0$,-1.51e+08),9.27e+07) →Sub(0.92,Pow(Add($x_0$,1.03),0.652))→Sub(Pow(Sub($x_0$,x[0]),x[1]),Pow(x[2],$x_0$)) |
| 687_sleuth_ex1605 | $x_4$→Mul($x_4$,0.774)→Mul(x[0],Mul($x_4$,x[1]))→Mul(Pow(Add($x_3$,2.71),-0.368),$x_4$) →Mul(Mul($x_4$,0.736),Pow(Pow($x_3$,-0.203),$x_4$)) |
| 690_visualizing_galaxy | $x_3$→Mul($x_1$,-0.894)→Sub(Mul($x_0$,0.279),$x_1$)→Mul($x_1$,Sub(Pow(0.586,$x_2$),1.77)) →Div(Sub(Mul($x_0$,0.303),$x_1$),Pow($x_3$,0.449)) |
| 706_sleuth_case1202 | $x_4$→Mul($x_4$,0.798)→Sub($x_4$,Mul($x_2$,0.279))→Sub(Mul($x_2$,-0.242),Mul($x_4$,-0.861)) →Div(Sub(Div($x_4$,0.283),$x_2$),Sub($x_2$,-4.13)) |
| 712_chscase_geyser1 | $x_1$→Mul($x_1$,0.877)→Sub(1.24,Pow(0.529,$x_1$))→Sub(Mul($x_1$,0.888),Div(0.0774,$x_0$)) →Sub(Sub(Pow($x_1$,-0.151),1.06),Mul($x_1$,-0.84)) |

## C. Classification of Dataset Type

Table 4: Classification of dataset type based on performance of SR algorithms with respect to the APO front.

| Type | Datasets |
|------|----------|
| Type I | 547_no2, 556_analcatdata_apnea2, 557_analcatdata_apnea1, 579_fri_c0_250_5, 594_fri_c2_100_5, 596_fri_c2_250_5, 597_fri_c2_500_5, 601_fri_c1_250_5, 611_fri_c3_100_5, 613_fri_c3_250_5, 617_fri_c3_500_5, 624_fri_c0_100_5, 631_fri_c1_500_5, 649_fri_c0_500_5, 656_fri_c1_100_5 |
| Type II | 1027_ESL, 522_pm10, 561_cpu, 663_rabe_266, 690_visualizing_galaxy |
| Type III | 210_cloud, 519_vinnie, 523_analcatdata_neavote, 678_visualizing_environmental, 712_chscase_geyser1 |
| Type IV | 1096_FacultySalaries, 192_vineyard, 228_elusage, 230_machine_cpu, 485_analcatdata_vehicle, 659_sleuth_ex1714, 665_sleuth_case2002, 687_sleuth_ex1605, 706_sleuth_case1202 |

## D. Validity of *K-expressions*

All *K-expressions* can be decoded into valid mathematical expressions. In Algorithm 1, $terminal\_symbol\_count$ is the length of the tail given by a formula, $head\_length$ is a hyperparameter that is the number of operators and operands in the head of a *K-expression*. For example, the *K-expression* '$* + -abcdef$' is decoded as $(a + b) * (c - d)$, where the length of the tail is 5 and the number of operators and operands in the head is 4.

In the main text, we claim the length of the tail is determined by $h \times (n_{max} - 1) + 1$, where $h$ is the head length and $n_{max}$ is the maximum operand arity of the primitive function set. The length of the tail is to ensure that there are no non-terminal symbols with empty arguments. It is computed from assuming the worst-case scenario where each symbol in the head is an operator, there will be at most $n_{max}$ arguments. We have $h$ symbols at the head, each with at most $n_{max}$ arguments to fill, but since each of the $h$ symbols fill up an empty spot, each symbol only creates at most $(n_{max} - 1)$ empty spots to fill. This leads to $h \times (n_{max} - 1)$ potential spots for the tail symbols to fill. Including the initial empty spot to be filled, we get $h \times (n_{max} - 1) + 1$.

## E. Extension to SRSD Metrics

SRSD (Matsubara et al., 2024) introduced a standard for metrics to use in SR analysis. For their accuracy metric, this is simply $R^2 > 0.999$ and can be obtained from the $R^2$ column. Of greater interest are the solution rate and NED introduced by SRSD. However, these are only computable on datasets with closed-form ground-truth, so they are not directly applicable to this work. Nonetheless, a meaningful way to incorporate these metrics is by taking the APO equations we found (see Appendix B) and treating them as 'proxy ground-truth', enabling a new metric to assess SR algorithms performance on black-box datasets.

## F. Verification on Physics Experiments

We also ran experiments on data from the Newtonian dynamics experiments by Cranmer et al. (2020) to demonstrate its applicability to recovering already verified equations for verification. Specifically, we applied the APO front on the data to recover the known force laws via the representations learned by the internal 'message function'. We used the 1-D version of orbital force equation and charged particle force equation in Cranmer et al. (2020). The ground truth equation structures appear on the APO front we find using our approach, and these equations are the true underlying physical law.

## G. Expanded Function Set

To perform a similar analysis that is extended to a much larger function set (i.e., {Add, Sub, Mul, Div, Pow, Sin, Cos, Arcsin, Arcos, Exp, Log, Max, Min}), our sampling of run-time estimates a $12.7\times$ increase in compute resources required. However, we recognize the interest in having some indication of expansion of the function set and have done so on a single random seed 860, and using only BFGS optimization in the supplementary materials.

## H. Regularization in SR

An alternative approach that we could have used is to add regularization and train on the test set, but it is not clear which regularization scheme or regularization parameters should be used to obtain the best upper bound. Thus, we opted to exclude regularization (e.g., length, complexity, parameter magnitudes) in obtaining the Pareto front.

From the supplementary materials given in the publicly available GitHub link, the relation between train and test $R^2$ is generally strong with little trade-off. However, the gap (e.g., train $R^2$ − test $R^2$) is more interesting and the supplementary materials data show some trade-off between the gap and model length, which warrants a more detailed analysis in future work.

## I. More Details on Two-mutation Analysis

In finding #3, we are interested in studying properties of the loss landscape that tell us the tendency of getting stuck at local optima and the difficulties in assessing the global optima. For typical machine learning, for loss landscapes similar to the Rastrigin function, in regions close to the global optimum, the function has many small depressions or 'basins of attraction' (valleys) that can trap an optimization algorithm in local minima. Although the global minimum exists, the surrounding parameter space is filled with numerous local minima that resemble shallow valleys. For SR, the loss landscape of greater interest is on the function structure rather than on the numerical parameters and 'considering if expressions that are two mutations away from the APO front are likely to mutate back into the APO front' is our way of assessing if the equations on the APO front are surrounded by many 'basins of attraction' that makes it tougher for SR algorithms to discover them.

## J. Operator Frequency Analysis

We can also analyze the frequency of operators appearing on the APO front to gain insights which can possibly inform hyperparameter settings in SR. For example, in evolutionary SR, a operator that appears with higher frequency on the APO front may benefit from having a higher chance of being selected with parts of an expression has to be filled at random (e.g., 1-mutation). From our data in the supplementary materials, for equations of length 9, the average frequency of Add, Sub, Mul, Div, Pow is 0.7000, 0.8625, 0.9500, 0.4833, 1.0042, respectively. Note that the frequency can exceed 1 because there can be more than 1 of the same operators in each equation.

## K. Suggestions for New APO Front-based Metrics

The APO plots allow users to come up with a large range of interesting new measures. A non-exhaustive list of suggested measures include: i). $R^2$ closeness, by measuring the percentage of datasets where the $R^2$ value obtained by the SR algorithm is within 0.1 (i.e., 0.1 vertical distance on the Pareto plot) of the extended APO front (plateau after max length searched), ii). Euclidian distance from a single algorithm with the front, though the relative magnitudes of axes need to be decided, iii). compare the entire field of SR with respect to the front with existing measures like hypervolume (HV). For the algorithms AFP, AFP_FE, AIFeynman, BSR, DSR, EPLEX, FEAT, FFX, GP-GOMEA, ITEA, MRGP, Operon, SBP-GP, gplearn, the values for $R^2$ closeness are 62%, 59%, 12%, 15%, 62%, 68%, 68%, 35%, 65%, 47%, 32%, 62%, 68%, 59%, respectively.

## L. Extended Plots

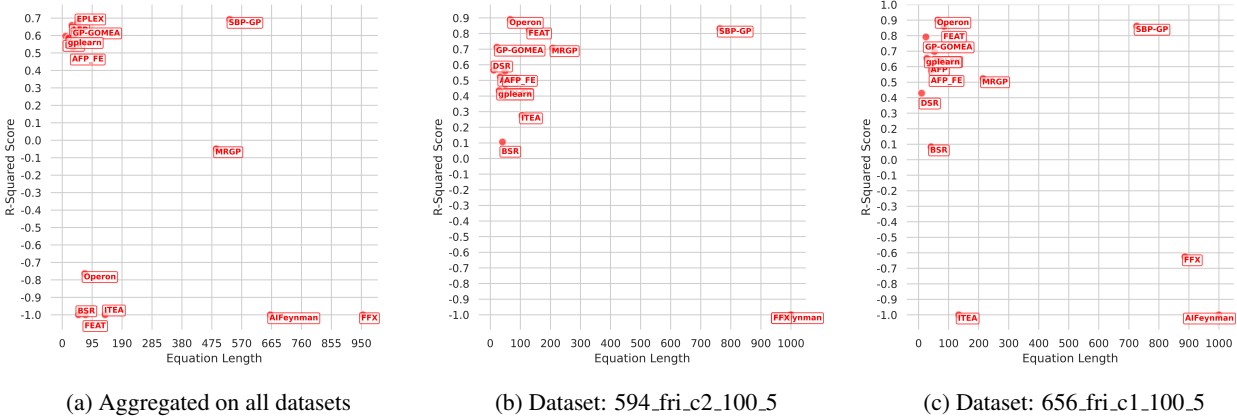

(a) Aggregated on all datasets      (b) Dataset: 594_fri_c2_100_5      (c) Dataset: 656_fri_c1_100_5

Figure 7: Plots from Figure 3 but with reduced truncation. Unlike rankings, using the actual metric values may be tricky to use in visualizing the results since the range of values can be large with a largely varying distribution.

## M. Worked Examples of Rank Inversion Paradox

Below are worked examples with exact values to illustrate, with a concrete example with real values, examples of when using rankings in Pareto analysis can lead to potentially contradictory results. We show an example where the aggregating method is to take the median of the ranks and another where the aggregating method is to take the average of the ranks.

### M.1. Using Median of Ranks

**Problem Setup:**

Consider 4 SR algorithms: A, B, C & D and 3 datasets, Ds1, Ds2, Ds3.

On Ds1, the raw performances are:

$R^2$ (higher is better) - C: 0.9, B: 0.8, A: 0.7, D: 0.6

Model size (lower is better) - B: 3, A: 5, C: 7, D: 9

On Ds2:

$R^2$ - C: 0.9, D: 0.8, A: 0.7, B: 0.6

Model size - A: 3, B: 5, C: 7, D: 9

On Ds3:

$R^2$ - B: 0.9, A: 0.8, C: 0.7, D: 0.6

Model size - D: 3, A: 5, B: 7, C: 9

Now, using the procedure in SRBench to determine which algorithms are relatively Pareto optimal (using SRBench github and consistent with Fig. 2 of the NeurIPS paper): first take the rank per dataset, then take the median of ranks.

**Case 1: Using only Algorithm A, B, C in Pareto analysis**

Here, we use the notation (3, 2) to represent that on Ds1, Algorithm A is Rank 3 in $R^2$, and Rank 2 in model size.

On Ds1:

A: (Rank 3 in $R^2$, Rank 2 in model size), B: (2, 1), C: (1, 3)

On Ds2:

A: (2, 1), B: (3, 2), C: (1, 3)

On Ds3:

A: (2, 1), B: (1, 2), C: (3, 3)

Median rank across Ds1, Ds2, Ds3:

A: (2, 1), B: (2, 2), C: (1, 3)

Thus, one would conclude Algorithm A & C are relatively Pareto optimal.

**Case 2: Using Algorithm A, B, C & D in Pareto analysis**

On Ds1:

A: (3, 2), B: (2, 1), C: (1, 3), D: (4, 4)

On Ds2:

A: (3, 1), B: (4, 2), C: (1, 3), D: (2, 4)

On Ds3:

A: (2, 2), B: (1, 3), C: (3, 4), D: (4, 1)

Median rank across Ds1, Ds2, Ds3:

A: (3, 2), B: (2, 2), C: (1, 3), D: (4, 4)

Thus, one would conclude Algorithm B & C are relatively Pareto optimal. Note that although only Algorithm D was added, Algorithm B is now suddenly optimal, and Algorithm A is suddenly not optimal. We can call this the 'Rank Inversion Paradox', inspired by (Chèze & Fieux, 2025; Zahir, 2009). Note that in taking actual quantities, any Algorithm that is not Pareto optimal will never be Pareto optimal with the addition of new Algorithms(s).

**M.2. Using Average of Ranks**

**Problem Setup:**

Consider 4 SR algorithms: A, B, C & D and 3 datasets, Ds1, Ds2, Ds3.

On Ds1, the raw performances are:

$R^2$ (higher is better) - A: 0.9, B: 0.8, C: 0.7, D: 0.6

Model size (lower is better) - C: 3, A: 5, D: 7, B: 9

On Ds2:

$R^2$ - A: 0.9, B: 0.8, C: 0.7, D: 0.6

Model size - B: 3, C: 5, A: 7, D: 9

On Ds3:

$R^2$ - C: 0.9, D: 0.8, B: 0.7, A: 0.6

Model size - B: 3, C: 5, A: 7, D: 9

Now, we first take the rank per dataset, then take the average of ranks.

**Case 1: Using only Algorithm A, B, C in Pareto analysis**

Here, we use the notation (1,2) to represent that on Ds1, Algorithm A is Rank 1 in $R^2$, and Rank 2 in model size.

On Ds1:

A: (Rank 1 in $R^2$, Rank 2 in model size), B: (2, 3), C: (3, 1)

On Ds2:

A: (1, 3), B: (2, 1), C: (3, 2)

On Ds3:

A: (3, 3), B: (2, 1), C: (1, 2)

Average rank across Ds1, Ds2, Ds3:

A: (1.67, 2.67), B: (2, 1.67), C: (2.33, 1.67)

Thus, one would conclude Algorithm A & B are relatively Pareto optimal.

**Case 2: Using Algorithm A, B, C & D in Pareto analysis**

On Ds1:

A: (1, 2), B: (2, 4), C: (3, 1), D: (4, 3)

On Ds2:

A: (1, 3), B: (2, 1), C: (3, 2), D: (4, 4)

On Ds3:

A: (4, 3), B: (3, 1), C: (1, 2), D: (2, 4)

Average rank across Ds1, Ds2, Ds3:

A: (2, 2.67), B: (2.33, 2), C: (2.33, 1.67), D: (3.33, 3.67)

Thus, one would conclude Algorithm A & C are relatively Pareto optimal. Note that although only Algorithm D was added, Algorithm C is now suddenly optimal, and Algorithm B is suddenly not optimal. We can call this the 'Rank Inversion Paradox', inspired by (Chèze & Fieux, 2025; Zahir, 2009). Note that in taking actual quantities, any algorithm that is not Pareto optimal will never be Pareto optimal with the addition of new algorithms(s).

## N. Other Complexity Measures

In the supplementary materials, we also include other complexity measures such as the count of operators, the count of numerical constants, Kommenda's complexity (Kommenda et al., 2015), Virgolin's trained linear elastic net interpretability estimator (we use their trained rescaled coefficients) (Virgolin et al., 2020) and Vladislavleva's order of non-linearity (using =1e-6 as done in their work) (Vladislavleva et al., 2008). We have also added Peterson's complexity that is used in DSR (Petersen et al., 2019).

