# OpenReview forum: "Pareto-Optimal Fronts for Benchmarking Symbolic Regression Algorithms"
_ICML.cc/2025/Conference — ICML 2025 poster_

### Official Review · Reviewer_GBr3 · 2025-03-08

**Overall Recommendation:** 3

**Summary:**

The authors aim to generate a set of "absolute pareto optimal" model results for 34 of the datasets in SRBench, as a way of having an upper bound on the performance of symbolic models on those datasets up to a given equation length. They exhaustively search for equations for those datasets up to a given length and use different numerical optimizers to tune the constants. They then compare to SRBench results and argue for a number of analysis conventions.

**Claims And Evidence:**

> Convention #1: For SR benchmarking, axes in performance trade-offs plots should be in terms of the actual quantity and not in terms of ranking.
>

Raw performance comparisons and performance rankings achieve different goals and answer different questions.

Raw performance metrics answer the question “how much different is the performance of alg A from alg B on dataset X?” and work well when comparing methods on a single dataset like in Figure 2. (Even there, though, the collapse of equation lengths in 2c) hides the fact that SBGP, ITEA, FFX, MRGP, and AI-Feynman produce models of different complexity, which is made apparent when comparing rankings in 2a). However, they bias the comparison of algs across datasets difficult because the distribution of values varies among problems.

In contrast, performance rankings are the more robust way to answer “how often does alg A outperform alg B over (many) datasets?” and that is primary question multi-dataset benchmarks like SRBench try to answer. It is also why rankings are the accepted standard for statistical comparisons of algorithms over multiple datasets. (e.g., Demšar, J. Statistical Comparisons of Classifiers over Multiple Data Sets. *Journal of Machine Learning Research* **2006**, *7* (Jan), 1–30.)

The problems the authors point out are that 1) rankings change when the set of algorithms change; and 2) rankings lose the magnitude of the changes. These points are true but they are, IMO, well-known. One shouldn’t compare rankings for different experiments or to assess magnitude differences; they are used because they are useful for benchmarking over many datasets and answering whether algorithm performances differ or not.

> Convention #2: Aggregating results across datasets can have contradictory conclusions, it should be supplemented with analysis on individual datasets to confirm the trend.
>

Ironically, the problem illustrated in Figure 3a)-3c) is caused by aggregating the *actual quantities* rather than the rankings, which contradicts convention #1 recommendation.

It also appears (judging by operon R2) that the authors are using the mean R2 over datasets, when they should use the median. R2 is unbounded on the negative end, so median scores would  give a more robust estimate of performance across datasets. An even more robust way to estimate relative performances of algorithms across multiple datasets would be to use rankings, which are not dataset-dependent like the R2 scores and equation lengths are.

This problem (mean aggregation of raw performance scores) also biases the results in Figure 1.

**Essential References Not Discussed:**

Prior work is sufficiently discussed.

**Experimental Designs Or Analyses:**

See methods & evaluation criteria.

**Methods And Evaluation Criteria:**

> In our experiments, we use a primitive function set of {Add,Sub,Mul,Div,Pow}
>

The operator set the authors used doesn’t match the set used in developing/benchmarking SR algorithms on SRBench. La Cava et al 2021 / SRBench allowed the operator set {+, −, ∗, /, sin, cos, arcsin, arccos, exp, log, pow, max, min}. So, strictly speaking, the absolute pareto optimal (APO) fronts generated by the authors are not pareto-optimal with respect to the task definition in the SRBench results. I would expect the APO front to be heavily influenced by the chosen operator set, and the exhaustive space of equations to be much larger than what the authors have generated. So it is not clear to me that the APO results are directly comparable to SRBench without some large caveats.

In section 3, the authors argue against aggregating results, but then in all results figures, results with equation lengths greater than 20 (sometimes 100) are collapsed together. In some cases this presentation skews the Pareto front interpretation since most of the benchmarked methods produce results that are $\geq$ 20. Isn’t this expressly against the recommendation?


pretty much all of the Type 1 datasets are Friedman datasets, which are synthetically generated from ground-truth analytical equations. (ref: Jerome H Friedman. Greedy function approximation: A gradient boosting machine. Annals of statistics, pages 1189–1232, 2001.) if the goal is to produce absolute pareto optimal front for those equations, presumably you would want to include the ground truth equation.

the authors don’t make it clear how they are handling train/test splits to compare to SRBench. if they are just training (model parameters) on the full datasets, the APO models will be biased relative to ones that could be possibly obtained if they followed train/test procedure of the benchmarked methods. i.e., even if a “optimal” model in some sense exists, it is not a given that one could possibly find it via finite sample training.

**Other Comments Or Suggestions:**

I would suggest the authors totally remove the analysis conventions suggestions. I think they are misguided and lack nuance.

I would also suggest the authors improve their discussion of limitations to incorporate the points above.

Finally I encourage the authors to discuss what types of measures an APO could provide within SRBench.

**Other Strengths And Weaknesses:**

Strengths: Having a set of pareto-optimal models for SRBench is, in general, a good idea, as it sets a measurable goal for these datasets.

Weaknesses: I think the authors work has some serious caveats that cause it to fall a bit short of delivering on that goal, including:

1) restriction of operator space in generating equations
2) apparent optimization of model constants on test data, which likely overestimates performance potential of APO
2) proposed analysis conventions which are misguided IMO
3) coverage of only a fraction of the datasets in SRBench

**Questions For Authors:**

- Is there a measure like distance to the optimal Pareto front that could viably be used?

- what is the likelihood that some of the APO models actually appear in the pareto fronts of population-based SR methods, and aren't selected because of finite sample limitations?

**Relation To Broader Scientific Literature:**

The contribution of the paper builds on SRBench and subsequent papers that use this benchmarking resource. Its main contribution is to specify an upper front of performance for a subset of the datasets in that benchmark. to my knowledge this hasn't been suggested previously.

**Theoretical Claims:**

n/a

---

> ### Author Rebuttal · Authors · 2025-04-01
>
> > General
>
> Please refer to (G.1).
> > Claims
>
> (D.1) We thank the reviewer for the suggestions for “Convention #1”. We will improve the clarity of our discussion by explicitly stating that our comments refer only to Pareto fronts/optimality and not statistical tests in (Demšar, 2006).
>
> The reviewer noted that “One shouldn’t compare rankings for different experiments or to assess magnitude differences”, and this is precisely the issue with the SRBench paper and accompanying code (in SRBench github master/postprocessing/blackbox_results.ipynb and Figure 2 of their NeurIPS paper) which is the motivation for our convention. We felt it was urgent to highlight this point, given its widespread adoption in SR papers. We appreciate the reviewer’s agreement on this matter and will incorporate the quoted phrasing to enhance clarity.
>
> Furthermore, in other fields where Pareto fronts/optimality is used, e.g., economics, engineering, operations, multi-objective optimization, raw performance metrics serves as the axes. To our knowledge, SRBench’s analysis is one of, if not the only, to deviate from this practice by using rankings. Other reasons to use raw performance for Pareto fronts/optimality is to ensure the results are “transferable” and comparable across different studies, which other reviewers highlighted. We apologize for the previous wording that suggested a disagreement with the reviewer. We will add these clarifications to ensure that our position–fully aligned with the reviewer–is accurately conveyed.
>
> (D.2) For “Convention #2”, the aggregation affects both raw quantities and ranking, both convention #1 and #2 needs to be applied simultaneously. We will include median plots in Appendix L for completeness. Per the convention, we actually don’t recommend either the mean or median. Finally, Figure 1 is meant for illustration purposes, which we will address in convention #2 that unaggregated results are required for a holistic assessment.
> > Methods And Evaluation
>
> (D.3) Actually, SRBench does not use a consistent operator set for SR algorithms according to their paper, code and results file (i.e., $\texttt{.feather}$ file). We found {Add,Sub,Mul,Div,Pow} to be commonly included in most SR algorithms. Thus, we used it so that distance away from the APO front can be attributed to search inefficiency rather than a lack of certain functions. We will make this clearer in the paper. However, we still create some new assets for the expanded function set (please see (C.3))
>
> (D.4) We thank the reviewer and will add the full untruncated plots in Appendix L, the ordering of the SR algorithms still remains the same under truncation, unlike taking average of the ranks as the axis for SRBench (which we recommend against), which can change the ordering depending on the algorithms selected.
>
> (D.5) We will include the data sampling process for the Friedman datasets in Appendix A.
>
> (D.6) In order to obtain the best upper bound on the performance with the best possible equation in the search space, we train on the full datasets. The alternative approach of using only the train set is problematic because we have to add regularization to the objective and it is not clear which to use to obtain the best upper bound. However, to provide more measurements to SR researchers we have included data on all combinations of optimization procedures (please refer to point (A.3) in the response to Reviewer NEMV). A meaningful future work enabled with the data is to explore which regularization techniques would create the closest performance to this upper bound on performance. We will include this discussion with plots in Appendix H.
> > Supplementary, Broader Literature, Essential References
>
> (D.7) We thank the reviewer for the validation.
> > Other Strengths And Weaknesses:
>
> (D.8) For the 4 numbered weaknesses please refer to points 1. (D.3), 2. (D.6), 3. (D.1) & (D.2), 4. (B.6), respectively.
> > Comments
>
> (D.9) For conventions, please refer to (D.1) & (D.2).
>
> (D.10) For our improvements in limitations, please see (A.1) & (C.3).
>
> (D.11) The APO plots allow users to come up with a large range of measures. A non-exhaustive list of suggested measures include: i). R-squared closeness, by measuring the percentage of datasets where the R-squared value obtained by the SR algorithm is within 0.1 (i.e, 0.1 vertical distance on the Pareto plot) of the extended APO front (plateau after max length searched), ii). Euclidian distance from a single algorithm with the front, though the relative magnitudes of axes need to be decided, iii). compare the entire field of SR with respect to the front with existing measures like IGD and HV. For the algorithms AFP, AFP_FE, AIFeynman, BSR, DSR, EPLEX, FEAT, FFX, GP-GOMEA, ITEA, MRGP, Operon, SBP-GP, gplearn, the values for i). are 62%, 59%, 12%, 15%, 62%, 68%, 68%, 35%, 65%, 47%, 32%, 62%, 68%, 59%, respectively. We will include the full results in Appendix K.
> > Questions
>
> (D.12) For both questions, please refer to point (D.11).

---

> > ### Comment · Reviewer_GBr3 · 2025-04-04
> >
> > I don't think the authors understood my critique of their proposed conventions:
> >
> > > The reviewer noted that “One shouldn’t compare rankings for different experiments or to assess magnitude differences”, and this is precisely the issue with the SRBench paper and accompanying code which is the motivation for our convention.
> >
> > By "experiments" I mean one shouldn't compare rankings between experiments _with different sets of algorithms_. One *should* use rankings or other problem-independent measures to compare the *same set of algorithms* across *multiple datasets*. This isn't just for statistical tests.
> >
> > > For “Convention #2”, the aggregation affects both raw quantities and ranking,
> >
> > Mean aggregation of R2 is extremely sensitive to outliers. Rankings don't have outliers, so it isn't true that they are affected the same way.
> >
> > > the ordering of the SR algorithms still remains the same under truncation, ...
> >
> > This isn't true, truncation introduces ties between algorithms that aren't there.
> >
> > > unlike taking average of the ranks as the axis for SRBench (which we recommend against), which can change the ordering depending on the algorithms selected.
> >
> > This also isn't true. If algorithm A is better than B, that relative ordering is maintained whether you compare A,B,C,D or A,B,C.
> >
> >
> > > (D.3) Actually, SRBench does not use a consistent operator set for SR algorithms according to their paper, code and results file
> >
> > The SRBench experiment defined a set of operators that could be used. Each algorithm, in turn, had different available operator implementations from that set. The authors' results here don't cover the possible set of operators, so their pareto optimality is not "absolute" w.r.t. the original design.
> >
> >
> > I would like reiterate my original point that rankings and R2 achieve different goals in benchmarking contexts and require a nuanced discussion.

---

> > > ### Author Response · Authors · 2025-04-05
> > >
> > > (D.13) We thank the reviewer for the feedback. We first present an example with concrete values to show how aggregate rankings work out. Consider 4 SR algorithms: A, B, C & D and 3 datasets, Ds1, Ds2, Ds3.
> > >
> > > On Ds1, the raw performance are:
> > >
> > > $R^2$ (higher is better) – C: 0.9, B: 0.8, A: 0.7, D: 0.6
> > >
> > > Model size (lower is better) – B: 3, A: 5, C: 7, D: 9
> > >
> > > On Ds2:
> > >
> > > $R^2$ – C: 0.9, D: 0.8, A: 0.7, B: 0.6
> > >
> > > Model size – A: 3, B: 5, C: 7, D: 9
> > >
> > > On Ds3:
> > >
> > > $R^2$ – B: 0.9, A: 0.8, C: 0.7, D: 0.6
> > >
> > > Model size – D: 3, A: 5, B: 7, C: 9
> > >
> > > Now, using the procedure in SRBench to determine which algorithms are relatively Pareto optimal (using SRBench github $\texttt{master/postprocessing/blackbox\\_results.ipynb}$ and consistent with Fig. 2 of their NeurIPS paper): first take the rank per dataset, then take the median of ranks.
> > >
> > > **Case 1: Using only Algo A, B, C**
> > >
> > > On Ds1:
> > >
> > > A: (Rank 3 in $R^2$, Rank 2 in model size), B: (2, 1), C: (1, 3)
> > >
> > > On Ds2:
> > >
> > > A: (2, 1), B: (3, 2), C: (1, 3)
> > >
> > > On Ds3:
> > >
> > > A: (2, 1), B: (1, 2), C: (3, 3)
> > >
> > > Median rank across Ds1, Ds2, Ds3:
> > >
> > > A: (2, 1), B: (2, 2), C: (1, 3)
> > >
> > > Thus, one would conclude Algo A & C are relatively Pareto optimal.
> > >
> > > **Case 2: Using Algo A, B, C & D**
> > >
> > > On Ds1:
> > >
> > > A: (3, 2), B: (2, 1), C: (1, 3), D: (4, 4)
> > >
> > > On Ds2:
> > >
> > > A: (3, 1), B: (4, 2), C: (1, 3), D: (2, 4)
> > >
> > > On Ds3:
> > >
> > > A: (2, 2), B: (1, 3), C: (3, 4), D: (4, 1)
> > >
> > > Median rank across Ds1, Ds2, Ds3:
> > >
> > > A: (3, 2), B: (2, 2), C: (1, 3), D: (4, 4)
> > >
> > > Thus, one would conclude Algo B & C are relatively Pareto optimal. **Note that although only Algo D was added, Algo B is now suddenly optimal and Algo A is suddenly not optimal.** We can call this the **“Rank Inversion Paradox”**, inspired by [1,2].
> > >
> > > (D.14) We thank the reviewer for the clarifications on “experiments”. By nature, the “set of algorithms” needs to be changed over time as new algorithms are developed and is not static. For instance, say in Year 2024, only 3 algo are available, Algo A, B, C (see (D.13)), the conclusion from using aggregate rankings is that Algo A & C are the best (i.e., relatively Pareto optimal). Then in Year 2025, Algo D (see (D.13)) is developed, which would necessitate its inclusion in the analysis (and hence a change in the set of algorithms analyzed), would yield the new conclusion from using aggregate rankings that Algo B & C are the best.
> > >
> > > (D.15) We agree that both raw quantities and ranking are not “affected the same way”, but rankings are still affected as seen in Example (D.13).
> > >
> > > (D.16) To clarify further, we will add the untruncated plots in Appendix L.
> > >
> > > (D.17) The observation that “If algorithm A is better than B, that relative ordering is maintained whether you compare A,B,C,D or A,B,C” is not true for aggregate ranks. We provide a counter-example in (D.13) above. Also, see [1,2].
> > >
> > > (D.18) In agreement with the reviewer, our understanding is also that each SR algorithm in SRBench has a different operator set. As addressed in (D.3), we will now include results on the larger set and provide rationale for why a smaller commonly used subset can be more useful.
> > >
> > > **(D.19) Appeal**
> > >
> > > We understand the reviewer disagrees with our convention which criticizes using aggregate rankings in Pareto analysis. We are not denying potential pros, but feel SR researchers should be made aware of the cons. An easy resolution to gain the reviewer’s complete support is to “totally remove the analysis conventions suggestions” as suggested, since the top contribution of this paper are the APO fronts anyway and not the conventions. However, we are very passionate about improving the field of SR and hope for one chance to win the reviewer over to our side before considering removal.
> > >
> > > i). We sincerely believe the conventions are necessary because the top-most priority is for easily transferable results across SR. If a new SR algorithm is developed, we want to be able to take the existing Pareto plot, and simply include a new coordinate and get consistent conclusions. The current approach of using aggregate rankings in Pareto plots means that existing plots cannot be reused and there is a potential for contradictory conclusions, as shown by the **“Rank Inversion Paradox”** in (D.13).
> > >
> > > ii). Most fields, if not all, do Pareto analysis in raw performance, not rankings. These fields include economics, engineering, operations, multi-objective optimization and “chip design and biomedical modeling” from Reviewer YJ4h.
> > >
> > > iii). The other reviewers support the conventions and even highlight similar issues in other fields to support our proposed conventions. This makes it awkward for us to remove the conventions without their opinion.
> > >
> > > **In the revision, we will include a nuanced discussion on pros and cons of using both raw quantities and ranking.**
> > >
> > > We hope if the remaining concerns have been addressed, the reviewer could consider increasing the score.
> > >
> > > [1] Chèze, G., et al. The Inversion Paradox and Ranking Methods in Tournaments.
> > >
> > > [2] “Rank reversals in decision-making”, Wikipedia

---

### Official Review · Reviewer_Gtam · 2025-03-14

**Overall Recommendation:** 4

**Summary:**

One common way to evaluate symbolic regression (SR) algorithms is to judge whether one method Pareto-dominates other SR algorithms. That means it has better performance for a given expression length. This paper proposes to evaluate SR methods against absolute Pareto-optimal solutions instead. It finds an absolute Pareto-optimal front of expressions for 34 real-world datasets from SRBench, a widely used SR benchmark. This is achieved by exhaustive search over all possible expressions of a given length and over eight different numerical optimization methods. The main contribution is a new baseline for benchmarking SR that informs SR researchers about the achievable limits of SR algorithms. Additionally, the paper proposes conventions for analyzing SR benchmark results and discusses several findings from the experiments.

## update after rebuttal
The current paper has certain limitations:
- not enough operators/functions included in the search space
- relatively small maximum length of the equations
- not all datasets in SRBench considered

However, I also do understand that some choices needed to be made to limit the already large computational resources needed.

Even in light of these limitations, I support the acceptance of this paper.

It is the first paper to provide some APO fronts, and I believe these can already be useful to researchers. From what I understand, the authors will release all evaluated equations, their fitness and complexity measured with respect to different metrics. These would be very interesting to analyze, especially for the datasets where there is a clear gap between the current methods and the found APO front. Very often, equations are only useful if they are "simple" enough to interpret. This paper shows that in some cases it is possible to find shorter equations with much better performance than the current methods (Figures 4c, 4d).

**Claims And Evidence:**

Claims in the papers are supported by clear and convincing evidence. The observations made about the current practices in SR literature are justified and their limitations discussed.

**Essential References Not Discussed:**

All essential references are discussed.

**Experimental Designs Or Analyses:**

The experimental designs and analyses seem valid. The paper describes the search space of equations and exhaustively searches through it. It also checks eight different numerical optimization algorithms and runs them for different seeds to ensure that they do not tend to be stuck in some suboptimal local minima. The random seeds are described and chosen to match the ones used in SRBench.

**Methods And Evaluation Criteria:**

The paper proposes a new baseline for a well-known and appropriate benchmarking dataset - SRBench. In particular, it focuses on the real-world datasets in SRBench which do not have a known ground truth. Thus establishing Pareto-optimal equations for these datasets is important.

**Other Comments Or Suggestions:**

None

**Other Strengths And Weaknesses:**

I think the paper contributes a very valuable asset to the community. Not only does it allow us to better judge the performance of SR algorithms in absolute terms, but it also gives us insights into the complexity of different datasets. I also believe that the dataset of all evaluated equations may provide additional insights, similar to the ones already present in the paper.

Table 1 (and Table 3) shows the APO front expressions for multiple datasets from SRBench. These are very interesting, especially in conjunction with Figure 4, which shows that for certain datasets, none of the tested methods is on the absolute Pareto front. This demonstrates a large gap in the capabilities of the current algorithms and hopefully will stimulate further research. Having "ground truth" equations for real-world datasets will allow for a better evaluation of how far we are from closing this gap.

The main weakness of the paper lies in a still relatively limited search space. I am fully aware that the current search space already requires extensive computational resources. However, it would be beneficial to understand what the Pareto front looks like for bigger equation lengths and for a richer function set (e.g., containing trigonometric functions or exponentials). It is possible that some of the methods are, in fact, on the absolute Pareto front for some datasets - we just do not know what this front looks like for larger equations. For instance, GP-GOMEA could, in theory, be optimal for datasets in Figures 4a and 4b. This is a significant limitation for datasets where the performance of the current algorithms is much higher than the maximum performance on the found absolute Pareto front (e.g., Figure 4a). I believe the limited search space (and thus unknown parts of the absolute Pareto front) should be emphasized in the limitations section.

**Questions For Authors:**

None

**Relation To Broader Scientific Literature:**

The paper addresses an important problem in SR benchmarking. Although the number of terms is currently the main measure used to judge the interpretability of an expression, it would be beneficial to briefly discuss other approaches to measuring equation's complexity (Vladislavleva et al., 2009; Vanneschi et al., 2010; Kommenda et al., 2015; Virgolin et al., 2020; Virgolin at al., 2021).

Kommenda, M., Beham, A., Affenzeller, M., and Kronberger, G. (2015). Complexity Measures for Multiobjective Symbolic Regression.

Vanneschi, L., Castelli, M., and Silva, S. (2010). Measuring bloat, overfitting and functional complexity in genetic programming.

Virgolin, M., De Lorenzo, A., Medvet, E., and Randone, F. (2020). Learning a Formula of Interpretability to Learn Interpretable Formulas.

Virgolin, M., De Lorenzo, A., Randone, F., Medvet,  E., and Wahde, M. (2021). Model learning with per
sonalized interpretability estimation (ML-PIE).

Vladislavleva, E. J., Smits, G. F., and den Hertog, D. (2009). Order of Nonlinearity as a Complexity Measure for Models Generated by Symbolic Regression via Pareto Genetic Programming.

**Theoretical Claims:**

N/A

---

> ### Author Rebuttal · Authors · 2025-04-01
>
> > General
>
> (G.1) We thank all 4 reviewers for their comprehensive reviews, identifying strengths of the work (e.g., “the authors have given concrete targets for future SR algorithm development”, “addresses a critical gap in SR benchmarking, where prior evaluations lacked universal reference point”) while providing actionable recommendations.
>
> Our personal motivation for this work began after receiving questions from SR researchers such as “is it still worth it to develop better SR algorithms?” and “is there still meaningful room for improvements in SR algorithms or has performance saturated?” in the context of new SR algorithms’ performance on SRBench being somewhat incremental (on average ~0.0077 $R^2$ improvements). Previously, one could only answer those questions subjectively based on trends, in the same spirit that Moore's Law does with speculating transistor density. With this work, we now confidently know that there are many cases in SRBench where there is still a large potential improvement. This is in spite of some computational limits we had to make given our already-large resource budget. Thus, this work is a data-driven justification that continuing research into developing new SR algorithms is still worth it.
>
> Though the main contribution is the Pareto front equations (i.e., given explicitly as equations in Appendix B and available in supplementary materials), the results also provide insights on how SR algorithms can achieve these performances. Particularly, it shows that search space of short, simple equations is sufficiently expressive and should be explored more in SR algorithms before expanding the search space to longer equations – a mechanism that is also related to increasing explainability and interpretability, which are the primary reasons for practitioners to pick SR over alternative machine learning algorithms in the first place.
>
> Among the various strengths, if you i). find that the Pareto fronts contributed in this paper are highly relevant and applicable to many, if not all, SR algorithms’ evaluation and ii). think that our work introduces and makes it convenient to add an informative and important baseline that you would like new SR algorithms and research papers to adopt, we hope that you could consider helping us advocate for acceptance of this work. This work makes publicly accessible, to communities of various funding levels, an important computationally-expensive baseline.
>
> We address the main concerns in the individual replies and hope that in light of these, the reviewers would consider increasing their recommendation.
>
> > Claims And Evidence, Methods And Evaluation, Experimental Designs, Supplementary
>
> (C.1) We thank the reviewer for the validation.
>
> > Broader Literature
>
> (C.2) We thank the reviewers for the suggestion to include more complexity measures to increase utility. Our provided data allows the computation of these with negligible costs as they are derived quantities from the obtained equation. In the csv files, we have extracted, for each row in the $\texttt{EquationStructure}$ column i). the count of operators, ii). the count of numerical constants, iii). Kommenda’s complexity, iv). Virgolin’s trained linear elastic net interpretability estimator (we use their trained rescaled coefficients), v). Vladislavleva’s order of non-linearity (using $\epsilon$=1e-6 as done in their work). We have also added Peterson’s complexity that is used in DSR. The files will be updated and the code for this processing will be made publicly available in the next upload opportunity. We are also committed to adding new metrics, and will update the files when new works such as [1] are publicly available. Below is an example (numbers truncated to 4 s.f.) of the improvement (e.g., $\texttt{1027\\_ESL\\_BFGS\\_3\\_860\\_summary.csv}$):
>
> |EquationLength|EquationStructure|...|OperatorCount|ConstantsCount|Kommenda|Virgolin|Vladislavleva|Peterson|...|
> |-|-|-|-|-|-|-|-|-|-|
> |...|
> |7|$\texttt{Sub(Mul(xdata[3],x[0]),Mul(xdata[1],x[1]))}$|...|3|2|6|0.7620|2|7|...|
>
> [1] Kacprzyk, K., & van der Schaar, M. (2025). Beyond Size-Based Metrics: Measuring Task-Specific Complexity in Symbolic Regression. In AISTATS.
>
> > Other Strengths And Weaknesses
>
> (C.3) We thank the reviewers for the suggestions for expansion of the function set. To perform a similar analysis, our sampling of run-time estimates a 12.7X increase in compute resources required. However, we recognize the interest in having some indication of expansion of the function set and have done so on a single random seed 860, and using only BFGS optimization, which will be included in supplementary materials available to readers and discussed in Appendix G. However, for the expansion of length, even a small increase in length led to an estimated 84.0X increase cost, so we are unable to provide that even with a single random seed and only BFGS. We will make these limitations clearer in the limitations section.

---

### Official Review · Reviewer_YJ4h · 2025-03-14

**Overall Recommendation:** 3

**Summary:**

This paper proposes an absolute evaluation criterion for symbolic regression, namely the Absolute Pareto Optimality (APO). At the same time, the effects of eight different optimization algorithms are analyzed. The establishment of this criterion is of great significance, as it provides a fair and reliable benchmark for researchers to evaluate the performance of symbolic regression algorithms, and solves the problem that in previous comparisons of algorithms, only relative indicators such as the recovery rate and R-squared value could be used for evaluation. The exhaustive search that consumed 1,480,000 supercomputer core-compute-hours demonstrates the author's determination. In this way, future comparisons of various methods will be more fair. It is expected to address the problem where one paper claims that its own method is excellent, but the results are different in another paper.

## update after rebuttal

I agree with the assertion from other reviewers about the attractiveness of the proposed method, and I also consider that there are some problems in this paper. The responses from authors address most of my concerns, even though they are not quite satisfactory. Since I acknowledge the value of the idea proposed by this paper, I am inclined to accept it and will keep my rating.

**Claims And Evidence:**

The authors claim three main contributions in this paper, and there are related evidences to support these claims:

1.They establish absolute Pareto-optimal fronts for 34 real-world datasets by exhaustively searching expressions up to fixed sizes using gene expression programming and K-expressions. This is rigorously supported by their computational effort and empirical comparisons showing gaps in current SR algorithms. The publicly released APO expressions and performance metrics provide a concrete, reproducible baseline.

2. The authors propose standardized conventions for SR benchmarking, advocating the use of actual metric values instead of rankings. This is validated through examples shown in Figure 2 and Figure 3, demonstrating how ranked axes distort conclusions. The proposal's logic aligns with the need for transferable and interpretable benchmarks.

3. The authors conduct an empirical comparison of numerical optimization methods for SR, showing minimal impact on APO front quality. Quantitative results and low $R^2$ variance across methods robustly support this claim.

**Essential References Not Discussed:**

I do not recognize missing essential references that are not discussed.

**Experimental Designs Or Analyses:**

The experimental design and result analysis in this paper are methodologically sound and rigorously structured. The authors construct absolute Pareto-optimal fronts through exhaustive search using gene expression programming and K-expressions across 34 real-world datasets. This approach aligns with multi-objective optimization principles and leverages prior validated techniques. Testing eight numerical optimization methods further ensures robustness, mirroring engineering practices where Pareto solutions are evaluated across diverse algorithms.

The experimental results are also analyzed rigorously. (1) Datasets are classified into four types, with some exposing insufficient exploration of compact expressions, which is a finding consistent with Pareto front "inaccessibility" observed in optimization studies. (2) Stability tests reveal minimal impact of numerical methods on APO front quality, demonstrating experimental control over confounding variables.

**Methods And Evaluation Criteria:**

The method proposed in this paper is important to the field of symbolic regression and its benchmarking practices.

1.By constructing absolute Pareto-optimal fronts through exhaustive search, the authors provide an objective, domain-agnostic baseline for SR algorithms. Unlike relative Pareto fronts, APO fronts define the theoretical performance ceiling for any SR method, enabling researchers to quantify how close their algorithms are to the "best possible" expressions for real-world datasets. This addresses a critical gap in SR benchmarking, where prior evaluations lacked universal reference points.

2.The APO fronts reveal that state-of-the-art SR algorithms systematically underperform on short expressions. This insight directs the SR community to prioritize improving compactness-aware search strategies, balancing accuracy and interpretability.

3.The proposal to use actual metrics ($R^2$, expression length) instead of rankings mitigates biases introduced by algorithm selection and enhances reproducibility. For example, Figure 2 demonstrates how ranked axes can misleadingly compress performance differences, while actual values expose true gaps. This standardization fosters fairer comparisons and accelerates progress by aligning the community on shared evaluation criteria.

**Other Comments Or Suggestions:**

I have a minor suggestion to refine the expression. In the abstract, there is a sentence "serves as an important benchmark that serves as a performance limit" --> “serves as an important benchmark and performance limit".

**Other Strengths And Weaknesses:**

Strengths

1.This paper builds on Pareto-optimality principles, proposing a framework for generating absolute Pareto fronts through exhaustive search. This aligns with multi-objective optimization methodologies. The use of gene expression programming and K-expressions ensures valid mathematical structures, reflecting prior work in symbolic regression and genetic algorithms.

2. The public release of all expressions, optimization parameters, and performance data reduces redundant computational efforts. The proposal to use actual metrics ($R^2$, expression length) over rankings addresses biases seen in multiagent learning evaluations.

Weakness:

While multi-seed experiments mitigate local optima risks, the “absoluteness” of APO fronts remains contingent on numerical optimization convergence. It is a limitation also noted in game-theoretic learning, where Pareto outcomes depend on assumptions like non-deceptive opponent behavior.

**Questions For Authors:**

1.How do you address potential limitations in achieving global optimality due to local minima in numerical optimization, despite using multiple random seeds? Are there theoretical or empirical guarantees that the APO front represents the global Pareto-optimal set for the expression spaces explored?

2.The APO fronts are derived from 34 SRBench datasets. How do you ensure these results generalize to domains with higher dimensionality, dynamic environments, or datasets outside SRBench?

3.How should SR researchers leverage the APO front to improve existing algorithms? For instance, would integrating APO-based heuristics enhance search efficiency?

4.Please explain the following terms: max_arity, terminal_symbol_count, head_length.

5.Why the length of the tail is determined by $h\times (n_{max}-1)+1$？

6.In Figure 5, why do different optimization methods need to be used to obtain optimal results for different data sets?

7.In Finding #3, why do you say you can describe the loss landscape by considering if expressions that are two mutations away from the APO front are likely to mutate back into the APO front?

**Relation To Broader Scientific Literature:**

The paper’s contributions are related to broader scientific literature:

1.The absolute Pareto-optimal front concept extends classical Pareto optimality principles, widely applied in fields like chip design and biomedical modeling. By exhaustively generating APO fronts for symbolic regression, the work mirrors deterministic Pareto methods while adapting them to data-driven modeling challenges, thereby bridging optimization theory and interpretable machine learning.

2.The empirical comparison of numerical optimization methods aligns with studies evaluating multi-objective algorithms, such as orthogonal evolutionary strategies or learning automata. Findings that numerical methods minimally impact APO quality resonate with literature emphasizing Pareto front stability under parameter variations, reinforcing the validity of SR algorithm evaluations.

**Theoretical Claims:**

I do not find theoretical claims in this paper.

---

> ### Author Rebuttal · Authors · 2025-04-01
>
> > General
>
> Please refer to point (G.1) in the response to Reviewer Gtam.
> > Claims, Methods and Evaluation, Experimental Designs, Supplementary, Broader Literature, References
>
> (B.1) We thank the reviewer for the validation.
>
> > Other Strengths And Weaknesses
>
> (B.2) We thank the reviewer for the reference to problems in using ranking in other fields such as multiagent learning evaluations. We will add this to the revision to link it to broader literature.
>
> (B.3) For “absoluteness”, please refer to point (A.1) in the response to Reviewer NEMV.
>
> > Other Comments
>
> (B.4) We thank the reviewer and will refine the sentence.
>
> > Questions For Authors
>
> (B.5) We used multiple random seeds and also multiple numerical optimization methods. We can guarantee all possible structures (within a certain complexity) are searched, but to the best of our knowledge, there are no theoretical or empirical guarantees in literature for the numerical optimization methods for all the different structures, so these are the best results we can produce with current technology and knowledge. We will be clearer on this in the revision as outlined in (A.1).
>
> (B.6) We thank the reviewer for the question. Because the main reasons to select SR in the first place is its explainable and interpretable models via short concise equations, when handling datasets with higher dimensions, feature selection can be performed first. Feature selection is also already a common preprocessing step in SR algorithms since SR algorithms do not scale well with dimensions. However, the best feature selection method to use for SR is highly debatable, hence we focused on datasets with low-dimensions. SRBench is already a compilation of datasets from multiple environments and can be said to be the de facto benchmark for SR, with extensively-tuned performance of SR algorithms, we did not use other datasets.
>
> (B.7) For improving existing algorithms, please see point (G.1) para 3. Additionally, using loss landscape analysis (“Finding #3”) but using k-mutations instead of 2-mutations, and finding the best k, can be incorporated into SR algorithms, where greedy k-mutations are performed for every candidate (i.e., for every candidate structure, greedily search the best k-mutations). Note that the second suggestion is an ongoing future work which needs more thorough analysis, but is only enabled by having the APO fronts.
>
> (B.8) max_arity is the number of arguments of the function with the most arguments, terminal_symbol_count is the length of the tail given by a formula, head_length is a hyperparameter that is the number of operators and operands in the head of a K-expression. We will include a longer explanation with simple examples of K-expression as well as proofs of the properties of K-expressions by Ferreira (2002) in Appendix D.
>
> (B.9) The length of the tail is to ensure that there are no non-terminal symbols with empty arguments. It is computed from assuming the worst-case scenario where each symbol in the head is an operator, there will be at most $n_{max}$ arguments. Since the symbol itself would fill an empty argument as well (with the exception of the first symbol), we have $h \times (n_{max}-1)+1$. We will include this in Appendix D.
>
> (B.10) We thank the reviewer for the question about Figure 5 in “Finding #2”. We did this because we wanted to show a variety of rarities of equations in the top-bin, so for each unique combination of i). random seed, ii). optimization method and iii). dataset, we made a histogram. The 3 in Figure 5 “were selected as their top-bin in the histogram had the minimum, median and maximum value among all other histograms”. As we learn later in “Finding #4”, no optimization method provides a clear prediction performance advantage. To communicate our findings more effectively in the revision, we will fix the histograms in Figure 5 to BFGS only, so as not the conflate the message of “Finding #2” with “Finding #5”.
>
> (B.11) For “Finding #3”, we are interested in studying properties of the loss landscape that tell us the tendency of getting stuck at local optima and the difficulties in assessing the global optima. For typical machine learning, for loss landscapes similar to the Rastrigin function, in regions close to the global optimum, the function has many small depressions or "basins of attraction" (valleys) that can trap an optimization algorithm in local minima. Although the global minimum exists, the surrounding parameter space is filled with numerous local minima that resemble shallow valleys. For SR, the loss landscape of greater interest is on the function structure rather than on the numerical parameters and “considering if expressions that are two mutations away from the APO front are likely to mutate back into the APO front”, is our way of assessing if the equations on the APO front are surrounded by many "basins of attraction" that makes it tougher for SR algorithms to discover them. We will include this discussion in Appendix I.

---

> > ### Comment · Reviewer_YJ4h · 2025-04-09
> >
> > I am basically satisfied with the authors' responses and will keep my rating.

---

> > > ### Author Response · Authors · 2025-04-09
> > >
> > > We thank the reviewer for the time and effort invested in this process.
> > >
> > > We hope to have the opportunity to present this work at ICML, where we believe it can both advance SR benchmarking and spark critical discussion to move benchmarking in SR toward a higher level of maturity comparable to that of the top ML subfields.

---

### Official Review · Reviewer_NEMV · 2025-03-14

**Overall Recommendation:** 3

**Summary:**

This paper proposes the absolute Pareto optimal (APO) front as a new benchmarking methodology for evaluating symbolic regression (SR) algorithms. Conventional SR evaluation is based on relative Pareto dominance with respect to other algorithms, but this does not provide any measure of efficiency or achievable limits. The authors established the theoretical limits of the trade-off between expression length and prediction performance (R-squared) through exhaustive search on 34 real-world datasets in SRBench. Specifically, they generated all possible expressions (within a specific length limit) using the K-expression format, and applied eight different numerical optimization methods to find the optimal parameters. As a result, the difference in performance between the current SR algorithm and the APO front was clarified, and it was shown that many algorithms were unable to find potential optimal solutions, especially in the search for short expressions. The paper also proposes new conventions for SR benchmark analysis (using actual values, and performing individual analysis rather than just aggregating results across datasets). The results of this research, which required a huge amount of computing resources (approximately 1.48 million core compute hours), are published as a valuable baseline for future SR algorithm development.

## Update After Rebuttal

After careful consideration of the authors' rebuttal, I am changing my evaluation from "weak reject (2/5)" to "weak accept (3/5)". The authors have sufficiently addressed my main concerns and promised the following improvements:

1. Clear explanation of the limitations of "absolute Pareto optimality" and improved notation (A.1)
1. Addition of complexity metrics beyond expression length (Kommenda, Virgolin, Vladislavleva, etc.) (A.3, C.2)
1. Provision of performance evaluations on both training and test data (A.3)
1. Self-contained explanation including proofs of K-expression properties (A.4)
1. Validation of the practical value of the APO front in modeling physical phenomena (A.11)
1. Provision of additional analyses such as function frequency analysis (A.14)

What particularly influenced my evaluation is that, as other Reviewers (especially Gtam) have emphasized, this research represents an important contribution to the evaluation of SR algorithms and provides a valuable resource to the research community despite its high computational cost. The improvements indicated by the authors address the limitations of the paper and enhance the practical utility of the results.

I also understand the concerns raised by other reviewers (particularly Reviewer GBr3) regarding the analysis conventions, but the authors' rebuttal (especially D.13-D.19) provides compelling examples of why traditional ranking-based analysis is problematic. On this point, while recognizing that there are various perspectives, I believe there is value in the practices proposed by the authors.

Overall, I am confident that this paper makes a valuable contribution to SR research, and with the implementation of the improvements promised by the authors, its value will be further enhanced. Therefore, I support the acceptance of this paper.

**Claims And Evidence:**

The arguments in this paper are well supported by the evidence presented. First, the argument regarding the construction of the APO front is supported by data obtained as a result of exhaustive search using a vast amount of computing resources. The authors explain the search method in detail and present it as a reproducible algorithm.

The authors' claims regarding the comparison of the current SR algorithm and APO fronts are supported by detailed analysis of 34 datasets, which are classified into four types to clearly explain the trends. In particular, Figure 4 visually shows representative examples of each type to support the claims.

The claims regarding the comparison of numerical optimization methods are also supported by specific data, such as the measurement of differences in distribution using KL divergence and statistics on the ratio of equations generated on the APO front.
However, there are some limitations to the claim of “absolute” Pareto optimality. As the authors themselves acknowledge, there is no guarantee that the numerical optimization methods used can find the true global optimum solution, and in this respect, it cannot be said to be truly “absolute”. However, efforts have been made to mitigate this problem by using multiple random number seeds and eight different optimization methods.

**Essential References Not Discussed:**

The paper cites and discusses a certain amount of relevant major literature. The major previous studies are appropriately cited in each category of symbolic regression benchmarking (SRBench), utilization of K-expression, exhaustive search SR, and numerical optimization in SR.

As a recent development in SR benchmarking, the paper also mentions SRBench++ (de Franca et al., 2024), but on the other hand, it does not follow benchmarks that propose new datasets and evaluation metrics, such as SRSD (Matsubara et al., 2024).

**Experimental Designs Or Analyses:**

The experimental design is sound overall, and the results are valid. The authors describe the experimental settings in detail, including the selection of random number seeds, the set of primitive functions used, and the numerical optimization method, in order to ensure reproducibility.

The method for extracting APO fronts is clearly defined, and the procedure of selecting the formula with the highest R-squared score for each formula length for each dataset is logical. The condition that longer formulas must perform better than shorter formulas is also reasonable.

Of particular note is that the authors compared eight different numerical optimization methods and showed that there was little difference in their performance. This provides valuable insight into the selection of numerical optimization methods in SR research.

The limitations of the experiment include the restriction on the length of the formula (head length = 3 and 4 only), the restriction on the function set used (only 5 binary operations), and the selection of a specific data set (less than 1000 data points, less than 10 features), but these are reasonable choices given the constraints on computing resources.

**Methods And Evaluation Criteria:**

The proposed method and evaluation criteria are appropriate for the SR benchmarking problem. Exhaustive search using fixed-length expressions with K-expression provides a reasonable trade-off between limiting the search space and guaranteeing the generation of valid formulas.

The Pareto front of expression length and R-squared values as evaluation criteria directly correspond to the essential goal of SR (balancing predictive performance and interpretability). In addition, the evaluation criteria proposed by the authors (using actual values rather than rankings, and performing individual analysis as well as dataset aggregation) appropriately point out the problems with current SR benchmarking.

However, the fact that they do not take into account model complexity indicators other than formula length (e.g. the type of mathematical operations or the number of numerical constants) and that they only consider the R-squared value of the training data and do not evaluate generalization performance are methodological limitations. However, the authors are aware of these limitations, and given the constraints on computing resources, this is a reasonable choice.

**Other Comments Or Suggestions:**

- By verifying the formula used to construct the APO front in actual application examples (e.g. modeling physical phenomena), it may be possible to further demonstrate its practical value.

- It may be beneficial to conduct research on an extended version of the APO front that includes a wider range of functions (e.g. trigonometric functions, exponential functions, etc.).

- By reformulating the problem as a multi-objective optimization that includes R-squared values for test data as well as training data, it may be possible to construct an APO front that also takes into account generalization performance.

- By performing feature analysis (such as what structures and functions are frequently used) on the equations on the APO front, it is possible to gain further insights for designing efficient SR algorithms.

- The methods used in this research have the potential to be applied to benchmarking interpretable machine learning methods other than symbolic regression, and it is also worth exploring this direction.

**Other Strengths And Weaknesses:**

**Strengths:**

・Originality: The introduction of the concept of absolute Pareto optimality into the evaluation of SR algorithms can be seen as an innovation in the evaluation paradigm.

・Practicality: By providing a baseline in the form of the APO front, the authors have given concrete targets for future SR algorithm development. This is particularly valuable for algorithm development that focuses on the search for short formulas.

・Contribution to the research community: By publishing results that require a huge amount of computing resources, the authors have reduced the computational burden on other researchers.

Thorough analysis: The SR is examined from multiple angles, including a comparison of eight numerical optimization methods and an analysis of the loss landscape around the APO front.

**Weaknesses:**

- Limited versatility: The applicability to problems with more complex relationships is limited due to the set of functions used and the restriction on the length of the expression.

- Computational cost issues: The proposed method requires a large amount of computational resources, making it difficult to extend to new datasets and function sets.

・Lack of practical guidance: Although the gap between the APO front and the current algorithm is shown, there are few specific algorithm design guidelines to fill the gap.

・Interpretability definition: Only the formula length is used as an interpretability indicator, and other aspects (such as conceptual simplicity) are not considered.

**Questions For Authors:**

I look forward to your response to the concerns I have raised above, but I have no further questions.

**Relation To Broader Scientific Literature:**

In terms of the evaluation method for symbolic regression, this study extends SRBench (La Cava et al., 2021). While SRBench was based on relative Pareto dominance, this study introduces the concept of absolute Pareto optimality.

The K-expression-based approach is based on Ferreira (2002)'s genome-phenome system as an approach that combines the advantages of genetic algorithms and genetic programming. It has also been shown to be related to other K-expression-based algorithms, such as DistilSR (Fong & Motani, 2023).

The comparison of numerical optimization methods contributes to the research on the optimization of numerical constants in SR (Kommenda et al., 2020; Chen et al., 2015). In particular, the fact that it compares methods other than the BFGS algorithm (frequently used in Biggio et al., 2021; Petersen et al., 2019) on a large scale is new.

This study also mentions SR applications in various fields, such as physics (Udrescu & Tegmark, 2020), materials science (Wang et al., 2019), engineering (Martinez-Gil & Chaves-Gonzalez, 2020), and healthcare (Christensen et al., 2022), showing its relevance to a wide range of scientific literature.

**Theoretical Claims:**

This paper is mainly an empirical study, and there are few theoretical assertions that require formal mathematical proof. The assertions regarding the properties of K-expressions (such as the fact that all K-expressions can be decoded into valid mathematical expressions) are based on the cited literature (Ferreira, 2002) and are not directly proven within the paper.

The claim about the complexity of the exhaustive search space (O(d^l), where d is the number of variables in the dataset and l is the length of the expression) is based on basic combinatorics and is considered to be correct.

The core claim that the APO front represents the performance limit of any SR algorithm is logically derived from the nature of exhaustive search, but is subject to the aforementioned constraint of the limits of numerical optimization.

---

> ### Author Rebuttal · Authors · 2025-04-01
>
> > General
>
> Please refer to point (G.1) in the response to Reviewer Gtam.
>
> > Claims And Evidence
>
> (A.1) We thank the reviewers for advice on our discussion on the subtleties with the term “absolute” in the limitations section that includes our mitigation strategies. To further improve the clarity, we will denote APO with a subscript that states the primitive function set used and the numerical optimization method (e.g., ${APO}_{(+,-,*,/,**,sin,cos,...),SLSQP}$) as a caveat. Despite not having a true global numerical optimizer, we are already able to find fronts that have a large performance gap with the equations found via existing SR algorithms.
>
> > Methods And Evaluation
>
> (A.2) For complexity indicators, please refer to point (C.2) in the response to Reviewer Gtam.
>
> (A.3) We implement the reviewers’ suggestions and include more data for SR research. In the $\texttt{Extracted\\_APO\\_Fronts}$ folder, instead of having only one column for $\texttt{EquationParameters}$, we now have 3 columns, each consisting of a vector of numerical constants optimized based on all data, train data only and test data only, respectively. For each of the 3 vectors of numerical constants, we also include their performance on all data, train data only and test data only. These will be made publicly available in the next upload opportunity. Below is an example (numbers truncated to 4 s.f.) of the improvement (e.g., $\texttt{1027\\_ESL\\_BFGS\\_3\\_860\\_summary.csv}$):
>
> |…|EquationStructure|EquationParametersAll|R2FitAllEvalAll|R2FitAllEvalTrain|R2FitAllEvalTest|…|
> |-|-|-|-|-|-|-|
> |...|
> |…|$\texttt{Sub(Mul(xdata[3],x[0]),Mul(xdata[1],x[1]))}$|[0.5503,-0.5307]|0.8056|0.8009|0.8172|…|[0.5713,-0.5118]|0.8052|0.8013|0.8136| …|[0.4931,-0.5881]|0.8016|0.7941|0.8216|...|
>
> |…|EquationParametersTrain|R2FitTrainEvalAll|R2FitTrainEvalTrain|R2FitTrainEvalTest|… |EquationParametersTest|R2FitTestEvalAll|R2FitTestEvalTrain|R2FitTestEvalTest|...|
> |-|-|-|-|-|-|-|-|-|-|-|
> |...|
> |…|[0.5713,-0.5118]|0.8052|0.8013|0.8136| …|[0.4931,-0.5881]|0.8016|0.7941|0.8216|...|
>
> > Theoretical Claims
>
> (A.4) We will make the paper more self-contained by including proofs of the properties from Ferreira in Appendix D.
>
> > Experimental Designs, Supplementary, Broader Literature
>
> (A.5) We thank the reviewer for the validation.
>
> > Essential References
>
> (A.6) We thank the reviewer and will add all SRSD metrics. For accuracy metric, this is simply $R^2>0.999$. For solution rate and NED, these are only computable on datasets with closed-form ground-truth, so not available. However, we thought that a meaningful way to incorporate these metrics is by taking the APO equations we found and treating them as “proxy ground-truth”, enabling a new metric to assess SR algorithms performance on black-box datasets. We thank the reviewer for inspiring this additional use of the APO equations and will add this and the discussion of other less commonly-used but high quality datasets in literature (e.g., SRSD) in Appendix E.
>
> > Other Strengths And Weaknesses
>
> (A.7) For “Limited versatility”, please refer to point (C.3) for more results and discussion.
>
> (A.8) For “Computational cost issues”, to the best of our knowledge, there is no empirical or theoretical alternative which could create a similarly universal baseline, so we felt the benefits far outweighed the cost. Our cost reducing strategy is to open-source these files so that it is a one-off cost instead.
>
> (A.9) For “Lack of practical guidance”, we will add guidance in Section 4 similar to point (G.1) para 3.
>
> (A.10) For “Interpretability definition", please refer to point (C.2) where we add other complexity metrics.
>
> > Other Comments
>
> (A.11) We thank the reviewer for the creative and impactful suggestion. In the revision, we will add Appendix F which shows that in [1] in the Newtonian dynamics experiments where SR is applied on the internal functions, the equation structures discovered by Cranmer et al. appear on the APO front we find using our approach, and these equations are the true underlying physical law.
>
> [1] Cranmer, M., et al. (2020). Discovering symbolic models from deep learning with inductive biases. NeurIPS.
>
> (A.12) For APO front on a wider range of functions, please refer to point (C.3).
>
> (A.13) We thank the reviewer for the idea. Using the data in point (A.3), we will include results of train vs test $R^2$ in Appendix H. In most cases, there is no trade-off, so we also include results of train minus test $R^2$ against length.
>
> (A.14) We thank the reviewer again and add function frequency analysis in Appendix J. For example, for equations of length 9, the average frequency of Add, Sub, Mul, Div, Pow is 0.7000, 0.8625, 0.9500, 0.4833, 1.0042, respectively, which can possibly inform the settings when generating the candidate solutions in SR algorithms.
>
> (A.15) We plan to explore using this approach for various permutations of common activation functions in small neural network architectures as a baseline for NAS algorithms.

---

### Decision · Program_Chairs · 2025-05-01

**Decision:**

Accept (poster)

**Comment:**

This submission received four reviews: 3 weak accept and 1 accept. The four reviewers all joined the internal post-rebuttal discussion.

While some of them are on the fence about the decision, they are inclined to accept this work. They value its analysis and attractive proposal of the absolute pareto-optimal fronts for the SRBench datasets. I sometimes find recent SR papers with ranking-based evaluations that a little bit misleading, specifically for top-ranked methods for each category, and I would like to welcome the proposed conventions.

Based on the internal discussion and reviews, I recommend that ICML 2025 accepts this work.
At the same time, I share non-trivial concerns from the reviewers that I hope the authors address in their camera-ready or their future studies:

> inability of the method to scale much larger than it currently does

> I would prefer to see more operators/functions included in the search space, a larger maximum length of the equations, and more datasets.

> it is not easy for other users to build upon the work. The authors answer my question about how to use their work to improve existing algorithms in (B.7), which is not helpful enough.

> The paper has several limitations including constraints on "absolute" Pareto optimality due to restricted operator sets, limitations on equation length, and coverage of only a subset of SRBench datasets. These limitations may restrict the applicability of the findings to broader symbolic regression contexts.

> coverage of only a fraction of the datasets in SRBench

> optimization of model constants on test data, which likely overestimates performance potential of APO

Lastly, I want the authors to make sure that the camera-ready has the following changes for future studies in the research community that follow this work.

- Publish the data used in this study (including analysis) and add the URL to the camera-ready as promised in the paper since some of the reviewers and I expect the assets to be published
- Publish code with clearer, self-contained instructions to reproduce the reported results. Based on the current materials, a few of the reviewers find it difficult for other users to build on this work, which should be resolved prior to the publication as this work proposes new conventions
- Make claims a little bit more specific, especially when referring to SRBench. Since this work uses a fraction of SRBench datasets, it should be clarified when the authors make claims involving the datasets. E.g., Figure 1 should explicitly say 'using 34 of 122 black-box datasets from SRBench" instead of "Our APO front with SR benchmark (SRBench (La Cava et al., 2021)) results". This is not limited to Figure 1 caption, and the authors should carefully check and modify the claims regarding SRBench accordingly.